# ViTSP: A Vision Language Models Guided Framework for Solving Large-Scale Traveling Salesman Problems

**Zhuoli Yin**[1], **Yi Ding**[2], **Reem Khir**[1], **Hua Cai**[1,3]

[1]Edwardson School of Industrial Engineering, Purdue University, USA
[2]Elmore Family School of Electrical and Computer Engineering, Purdue University, USA
[3]School of Sustainability Engineering and Environmental Engineering, Purdue University, USA
`{zhuoliyin, yiding, rkhir, huacai}@purdue.edu`

## Abstract

Solving the Traveling Salesman Problem (TSP) is NP-hard yet fundamental for a wide range of real-world applications. Classical exact methods face challenges in scaling, and heuristic methods often require domain-specific parameter calibration. While learning-based approaches have shown promise, they suffer from poor generalization and limited scalability due to fixed training data. This work proposes *ViTSP*, a novel framework that leverages pre-trained vision language models (VLMs) to visually guide the solution process for large-scale TSPs. The VLMs function to identify promising small-scale subproblems from a visualized TSP instance, which are then efficiently optimized using an off-the-shelf solver to improve the global solution. *ViTSP* bypasses the dedicated model training at the user end while maintaining effectiveness across diverse instances. Experiments on real-world TSP instances ranging from 1k to 88k nodes demonstrate that *ViTSP* consistently achieves solutions with average optimality gaps of 0.24%, outperforming existing learning-based methods. Under the same runtime budget, it surpasses the best-performing heuristic solver, LKH-3, by reducing its gaps by 3.57% to 100%, particularly on very-large-scale instances with more than 10k nodes. Our framework offers a new perspective in hybridizing pre-trained generative models and operations research solvers in solving combinatorial optimization problems. The framework holds potential for integration into more complex real-world logistics systems. The code is available at `https://github.itap.purdue.edu/uSMART/ViTSP_ICLR2026`.

## 1 Introduction

The Traveling Salesman Problem (TSP) is a fundamental combinatorial optimization (CO) problem with broad real-world applications, including transportation, logistics, and chip design (Applegate, 2006; Yin et al., 2023). Efficiently solving TSPs not only yields economical and societal benefits across those domains but also informs the development of solution strategies for other CO problems. The operations research (OR) community has developed numerous exact and heuristic algorithms to address this NP-hard problem (Davendra, 2010). However, exact methods often **struggle to produce high-quality solutions as the problem size increases**. Heuristic algorithms offer faster approximate solutions, yet their effectiveness **depends on domain-specific knowledge and careful calibration of instance-specific parameters** (Adenso-Díaz & Laguna, 2006).

Advances of machine learning (ML) have led to various learning-based approaches for solving TSPs (referred to as *neural solvers*), including end-to-end models for solution construction (Vaswani et al., 2017; Jin et al., 2023; Sun & Yang, 2023; Li et al., 2024) and learned neural strategies for local improvement (Zong et al., 2022; Cheng et al., 2023; Ye et al., 2023; 2024b; Zheng et al., 2024). These methods shorten the computation time and maintain good solutions for in-distribution, small-scale instances (nodes < 1,000) (Wu et al., 2024). **However, they suffer from poor generalization and limited scalability as soon as the real-world problem deviates from the training data**.

The surge of pre-trained large language models (LLMs) and vision language models (VLMs) has raised interest in their potential for tackling optimization problems. Existing efforts mainly focused on end-to-end construction of text-based solutions (Yang et al., 2023; Elhenawy et al., 2024) or on heuristic designs (Ye et al., 2024a; Liu et al., 2024) that rely solely on textual information of TSP instances. While these studies open new perspectives on using generative models to rethink optimization, **their approaches fall short of demonstrating reliable performance on large-scale practical TSPs** (Khan & Hamad, 2024)

In this study, we reconsider how pre-trained generative models can effectively complement established OR techniques for solving large-scale TSP instances with varying distributions and scales, enabling broader optimization applications. We leverage pre-trained VLMs to provide adaptive decomposition heuristics that complement the existing optimization routine, as effective decomposition must account not only for spatial locality but also for combinatorial neighborhoods that help escape local optima. VLMs are well-suited for this task, as they can interpret instance-specific spatial structures by treating TSP instances as 2D images, enabling more informed selection of subproblems. Unlike ML approaches that require domain-specific training for graph embeddings, VLMs offer generic reasoning capabilities without costly data collection or retraining. The resulting subproblems, being smaller than their original TSP, can be reliably solved by exact solvers, avoiding performance degradation often experienced in learned neural solvers (Joshi et al., 2022; Wu et al., 2024).

We propose *ViTSP*, a Vision-guided framework for solving large-scale TSP. In *ViTSP*, VLMs guide the optimization process by identifying meaningful subproblems from visualized TSP instances, while an off-the-shelf solver continuously refines those subproblems. *ViTSP* orchestrates the two modules asynchronously to accommodate the heterogeneous computational overheads of input/output (I/O) intensive VLMs and CPU-intensive solvers. On unseen TSPLIB (Reinelt, 1991) instances ranging from 1k to 88k, *ViTSP* finds the global optimum in 11 out of 33 instances and outperforms baseline learning-based methods, whose performance degrades significantly compared to their reported in-distribution results. Compared to the best-performing heuristic solver, LKH-3, *ViTSP* converges to superior solutions under the same time budget and reduces optimality gaps by 12% to 100%. Our key contributions in this study are summarized below:

1. We propose a vision-guided solution framework *ViTSP* that hybridizes VLMs and off-the-shelf solvers to reach strong performance. *ViTSP* is able to adapt to TSP instances with varying distributions and scales.
2. Our approach leverages pre-trained VLMs to visually derive decomposition heuristics while bypassing the costly and time-consuming training and data curation for edge deployment.
3. We conduct experiments on (very-)large-scale instances to validate the effectiveness of *ViTSP*. The ablation studies further underpinned *ViTSP*'s ability to perform principled guidance. To the best of our knowledge, our work presents one of the most comprehensive evaluations of real-world TSPLIB instances with $N > 1000$, whereas few prior works reported sufficient results at this scale.

## 2 RELATED WORKS

Existing approaches of solving large-scale TSP can be categorized into three primary schemes: (1) OR approaches, (2) learning-based approaches, and (3) LLM/VLM-based approaches. We briefly review these works in this section, and we supplement the detailed discussion in the Appendix B.

### 2.1 OR APPROACHES

Exact algorithms typically require explicit mathematical formulations and search for exact solutions via branch and bound procedures (Laporte, 1992; Wolsey, 2020). Off-the-shelf exact solvers, such as Concorde, Gurobi, and OR-Tools, have the potential to reach global optimality. Among them, Concorde remains the state-of-the-art (SOTA), using specialized rules to speed up the search process. However, the computation time of exact solvers becomes intractable as the problem size increases.

Heuristic algorithms, such as farthest insertion (Rosenkrantz et al., 1974), genetic algorithm (Holland, 1992), and Lin-Kernighan-Helsgaun-3 (LKH-3) (Helsgaun, 2017), iteratively refine solutions based on hand-crafted rules. LKH-3 is regarded as the SOTA in solving TSPs. However, LKH-3 relies on tunable parameters, such as the number of total runs and candidate edges. Without domain knowledge and instance-specific calibration, achieving strong performance is often non-trivial. According to

Adenso-Díaz & Laguna (2006), only about 10% of the effort in developing and testing heuristics or metaheuristics goes into designing it, with the remaining 90% spent in parameter tuning.

## 2.2 LEARNING-BASED APPROACHES

Learning-based approaches for solving CO problems have gained wide attention since the surge of deep learning. These works commonly employ graph neural networks to embed TSP instances. The networks are trained using either supervised learning, which requires high-quality solutions from exact or heuristic methods as labels, or reinforcement learning, which relies on extensive trial-and-error (Fu et al., 2021). Existing works mainly deploy trained networks under two paradigms.

**End-to-end construction.** This paradigm seeks to learn a policy to directly construct a solution, using either autoregressive or non-autoregressive (heatmap-based) schemes. The autoregressive scheme trains attention-based neural networks (Vaswani et al., 2017; Kwon et al., 2020; Jin et al., 2023). The network sequentially constructs solutions by outputting one node at a time, with previous outputs incorporated into the network to guide the generation of subsequent nodes (Deudon et al., 2018; Kool et al., 2019). In contrast, the non-autoregressive approaches, such as Qiu et al. (2022); Sun & Yang (2023); Li et al. (2023; 2024), estimate the likelihood of connecting each edge between nodes and construct the solution in one shot.

**Local improvement.** This paradigm iteratively updates solutions using learned policies in two ways. First, it repeatedly selects partial problems or decomposes the whole problem into separate subproblems, and reconstructs them using a separate neural solver or an OR solver (Li et al., 2021; Fu et al., 2021; Zong et al., 2022; Cheng et al., 2023; Pan et al., 2023; Ye et al., 2024b; Zheng et al., 2024). Second, it learns to predict stepwise searching to assist existing OR algorithms (Xin et al., 2021; Hudson et al., 2022; Zheng et al., 2022; Wu et al., 2022; Ye et al., 2023; Ma et al., 2023).

Despite promising in-distribution performance presented, specialized learning-based approaches often fall short in handling out-of-distribution (OOD) instances since their neural networks were trained on fixed datasets (Li & Zhang, 2025). Therefore, they often fail to compete with the reliability of established OR solvers. Such limitations hinder their applicability at a practical scale. In fact, few studies have evaluated OOD performance on open-source TSPLIB instances with more than 5,000 nodes (Reinelt, 1991), limiting our understanding of their robustness in real-world scenarios.

## 2.3 LLMS/VLMS-BASED APPROACHES

The surge of pre-trained LLMs/VLMs has drawn wide attention for optimization problems, including TSP. Yang et al. (2023) treats node coordinates as text input and prompts LLMs to output solutions, but the resulting solutions exhibit large optimality gaps even on small instances (i.e., N = 50). Another approach in Elhenawy et al. (2024) uses TSP images and relies on the VLMs to read node indices to construct tours. This design has limited scalability, as densely distributed nodes make it difficult for the VLMs to correctly recognize node indices. Liu et al. (2024) prompts LLMs for automatic heuristics design and translates them into code. However, their text-only framework overlooks instance-specific spatial structure. Consequently, their standalone strategy exhibits large variance in optimality gaps when applied to varying TSPs, limiting their reliability for practical use.

## 3 METHODS

We reposition VLMs from unreliable end-to-end solvers to practical complements that can be integrated with established OR tools in building scalable optimization routines. In contrast to graph-based neural solvers, which demand extensive training and still struggle to generalize, we leverage the generic multi-modal reasoning of VLMs to process TSP as an image, enabling them to interpret spatial structures and provide adaptive decompositions without task-specific training.

Building on this motivation, we propose the *ViTSP* framework (Figure 1), which integrates VLM guidance into the optimization pipeline through three key modules: solution initialization, visual selection, and subproblem optimization. Starting from an initial solution for a given TSP instance (Sec 3.2 ), VLMs identify box coordinates that delineate promising subproblems for further refinement based on the visualized TSP solution (Sec 3.3. The exact solver then iteratively optimizes returned subproblems to improve global solutions (Sec 3.4). Iteratively solving subproblems allows certain subproblems to be optimized combinatorially under varying neighborhoods to escape local optima.

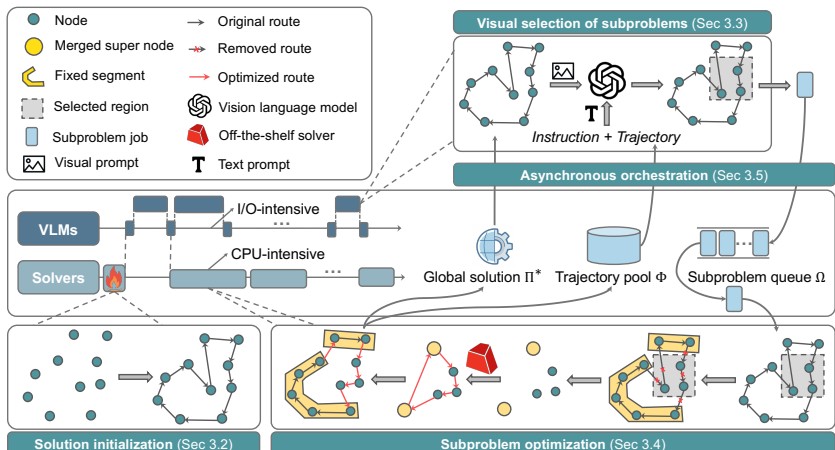

Figure 1: The vision-guided framework (*ViTSP*) for large-scale TSP, where pre-trained VLMs and off-the-shelf solvers are asynchronously coordinated to identify and optimize subproblems, respectively.

Since visual selection and subproblem optimization have distinct computational overheads, *ViTSP* coordinates their outputs asynchronously via a shared global solution, trajectory history, and subproblem queue to minimize the idle time in the subproblem optimization (Sec 3.5).

The key advantages that ensure the effectiveness and scalability of our approach are threefold:

1. We leverage the pre-trained models to provide a decomposition-like heuristic rather than an error-prone end-to-end solution construction. As strong generalists for user-specified tasks, these models eliminate the need for *ad hoc* (re-)training during real-world deployment.
2. Visually guiding the selection of box regions enables scalable subproblem decomposition based on the geometric structure even as the TSP instance grows in size.
3. We reformulate the identified subproblems as standard TSPs. This allows us to harness the robust exact solvers, guaranteeing high-quality improvement to the global solution.

## 3.1 PRELIMINARY: TRAVELING SALESMAN PROBLEM (TSP)

We briefly introduce the TSP in this section and provide detailed notations and descriptions in the Appendix A. A TSP is characterized by a list of nodes and the corresponding coordinate sets or the distance matrix. The goal of TSP is to find an optimal tour $\Pi^*$ that departs from an initial node, visits each node exactly once, and returns to the starting node, which minimizes the total distance traveled $L(\Pi^*)$. Notably, when the distance between two nodes is identical in both directions, the problem is known as symmetric TSP (**STSP**). In contrast, asymmetric TSP (**ATSP**) allows for different distances between certain node pairs in opposite directions.

## 3.2 SOLUTION INITIALIZATION

The *ViTSP* is warm-started using heuristic solvers. Critically, to effectively handle OOD instances, *ViTSP* avoids extensive parameter tuning and instead always uses the default settings of the solver. This eliminates the dependency on prior domain knowledge that would otherwise hinder *ViTSP*'s adaptability to varying instances.

## 3.3 VISUAL SELECTION OF SUBPROBLEMS BY VLMS

In the visual selection module $F_{\text{selector}}(\cdot)$, we prompt VLMs to select box regions and then formulate them as subproblems. We provide an overview of multimodal prompts and expected outputs in this section, and defer the detailed example prompts in Appendix C. The pseudocode for the visual selection process is detailed in Algorithm 1 in Appendix D.

**Visual prompts.** Given a TSP instance, we plot its nodes and their current connections on an image based on their 2D coordinates and the global solution $\Pi$. This image is input to VLMs as a visual prompt. An example of such visual input is illustrated in Figure 4 of Appendix C.

**Textual prompts.** We specify three types of information as textual inputs to an VLM: (1) *meta-instructions $I$*, detailing the subproblem selection task description and the expected output format; (2) *selection trajectories* $\Phi = \{\phi_1, \phi_2, ...\}$, served as memory to address the stateless nature of API-based VLM calls. Each trajectory entry $\phi_i$ includes a selected subproblem, the number of nodes within this subproblem, the solution gain through optimization, and the solver's runtime. The selection trajectories from earlier steps reveal instance-specific structures as the solving progresses, which informs VLMs to make better subsequent selections (Yang et al., 2023; Laskin et al., 2023; Monea et al., 2024; Moeini et al., 2025); (3) *Pending subproblems* $\Omega = \{\omega_1, \omega_2, ...\}$, indicating identified yet unsolved subproblems that remain in the queue. This avoids duplicated subproblems selected by VLMs.

**Image-level output.** The VLM is prompted to generate a quadruple $C = (x_{\min}, x_{\max}, y_{\min}, y_{\max})$ as a textual response. This quadruple represents the coordinates of a box region at the image level. As generative models, VLMs can be flexibly tailored to generate $Q$ coordinate sets $C_1, C_2, \ldots, C_Q$ per response (where $Q \geq 2$). We will leverage these multiple-subproblem outputs during module orchestration as described in Section 3.5.

**Forming a subproblem.** Given the current global solution $\Pi$, connections to the covered nodes within a given box region are removed, leading to a list of free nodes $W = \{w_1, w_2, ..., w_{|W|}\}$. The remaining connected nodes outside of the box form segments $K = \{(\pi_1^1, \ldots, \pi_{c_1}^1), \ldots, (\pi_1^{|K|}, \ldots, \pi_{c_{|K|}}^{|K|})\}$, where $(\pi_1^k, \ldots, \pi_{c_k}^k)$ denotes the $k$-th segment containing $|c_k|$ connected nodes; and $\pi_1^k$ and $\pi_{c_k}^k$ denote the starting and ending nodes in the $k$-th segment, respectively. As a result, the visual selection module produces a subproblem $\omega = (W, K)$.

**Zoom-in reselection.** *ViTSP* employs a zoom-in reselection to ensure scalability on very large-scale TSPs. Because such instances often exhibit highly dense node distributions, making connections in the initial visualization less discernible. To address this, a second round of selection is performed on the initially identified subregion bounded by $C$. If the number of nodes $|W|$ covered by the subregion exceeds a predefined threshold $\alpha$, the VLM zooms into it to examine the finer-grained pattern and identify a new quadruple $C'$.

## 3.4 SUBPROBLEM OPTIMIZATION

Rather than training a dedicated neural solver to optimize selected subproblems separately, as in prior works that adopt the local improvement paradigm (Cheng et al., 2023; Pan et al., 2023; Zheng et al., 2024), we transform the formulated subproblem $\omega = (W, K)$ into a standard symmetric TSP (STSP). As existing exact solvers are primarily designed for STSP, this reformulation allows us to leverage solvers to obtain a globally feasible solution with guaranteed quality.

### 3.4.1 REFORMULATING SUBPROBLEMS

During subproblem optimization, free nodes are reconnected either to other free nodes or to existing segments. Similarly, connections within each segment are preserved from the solution $\Pi$, while the links between the segment's endpoints and the rest of the segments or free nodes are refined. By aggregating each segment $(\pi_1^k, \ldots, \pi_{c_k}^k)$ into a super node $s_k$, we construct a new list of nodes of size $|K| + |W|$, denoted as $\{s_1, ..., s_{|K|}, w_1, ..., w_{|W|}\}$. This updated node list leads to a partially asymmetric TSP (ATSP), characterized by an asymmetric block distance matrix:

$$D_{ATSP} = \left[ \begin{array}{c|c} D_{|K|\times|K|} & D_{|K|\times|W|} \\ \hline D_{|W|\times|K|} & D_{|W|\times|W|} \end{array} \right]_{(|W|+|K|)\times(|W|+|K|)}$$

where $D_{|W|\times|W|}$ contains symmetric distances $d_{w_i,w_j}$ between free nodes. The submatrices $D_{|W|\times|K|}, D_{|K|\times|W|}$, and $D_{|K|\times|K|}$ are the root of asymmetry. $D_{|W|\times|K|}$ contains distances $d_{w_i,\pi_1^k}$ from free nodes to the starting nodes of fixed segments, whereas $D_{|K|\times|W|}$ represents distances $d_{\pi_{c_k}^k,w_i}$ from the ending nodes of fixed segments to free nodes; $D_{|K|\times|K|}$ indicates the distances $d_{\pi_{c_k}^k,\pi_1^k}$ from the ending nodes of fixed segments to the starting nodes of other segments.

We further transform this partially ATSP into a standard STSP to make it compatible with the solver. Following the approaches in Jonker & Volgenant (1983); Cirasella et al. (2001), the transformation introduces a dummy node $s_k'$ for each node $s_k$ in the ATSP, expanding the node set to $\{s_1, ..., s_{|K|}, s_1', ..., s_{|K|}', w_1, ..., w_{|M|}\}$. The resulting STSP is characterized by a symmetric block

distance matrix:

$$D_{STSP} = \begin{bmatrix} \infty & D_{|W|\times|K|}^T & \hat{D}_{|K|\times|K|}^T \\ D_{|W|\times|K|} & D_{|W|\times|W|} & D_{|K|\times|W|}^T \\ \hat{D}_{|K|\times|K|} & D_{|K|\times|W|} & \infty \end{bmatrix}_{(|W|+2|K|)\times(|W|+2|K|)}$$

where the diagonal of $\hat{D}_{|K|\times|K|}$ is set to be a small enough value compared to the original $D_{|K|\times|K|}$, which encourages the super nodes $k$ and their corresponding dummy nodes $k + |W|$ to be adjacently connected during the optimization.

### 3.4.2 SOLVING AND RECOVERING THE SOLUTION FOR THE ORIGINAL TSP

The solver optimizes an STSP using its $D_{STSP}$ and produces an optimal solution $\Pi^*_{STSP}$. The output solution $\Pi^*_{STSP}$ is then recovered into the corresponding ATSP solution $\Pi^*_{ATSP}$ by directly removing all dummy nodes in the solution $\Pi^*_{STSP}$. Furthermore, each super node $s_k \in \Pi^*_{ATSP}$ is unfolded into its original segment $(\pi^k_1, \ldots, \pi^k_{c_k})$. This recovery process results in an updated solution for the original TSP conditioned on the identified subproblem $\omega$: $\Pi^* = F_{solver}(D_{STSP}, T_{max} \mid \omega)$, where $T_{max}$ is the runtime limit set for the exact solvers. The time limit $T_{max}$ forces the solver to stop improving lower bounds and return the best incumbent solutions. This prevents the solver from getting stuck on certain subproblems for an excessively long time. We use the hill-climbing rule in accepting this new solution if it reaches a lower objective value than the current solution.

### 3.5 ASYNCHRONOUS ORCHESTRATION

The visual selection module $F_{selector}(\cdot)$ is I/O intensive, dominated by waiting for responses from the VLM server, whereas the exact solver module $F_{solver}(\cdot)$ is CPU-intensive. Due to their distinct computational profiles, sequential execution easily leaves solvers idle while waiting for VLM selection, and vice versa. To address this, *ViTSP* executes the optimization and selection modules asynchronously on multi-core CPU systems, assigning them to separate cores and coordinating through three shared components: global solution $\Pi$, trajectory pool $\Phi$, and subproblem queue $\Omega$. These components provide the necessary contextual information required for module execution.

To further improve efficiency, *ViTSP* deploys multiple VLMs and solvers. On the selection side, we employ both fast-thinking and reasoning VLMs, leveraging pre-trained models with complementary strengths (Shen et al., 2023; Snell et al., 2024; Kumar et al., 2025). Each single VLM is elicited to generate $Q$ coordinate sets $\{C_1, C_2, ..., C_Q\}$ per prompt, where $Q \geq 2$.

On the optimization side, multiple identical solvers retrieve and optimize subproblems from the shared queue in parallel, ensuring that newly generated subproblems are not left unprocessed. To mitigate conflicts in updating global solutions, *ViTSP* assigns $P$ "slave solvers" to optimize and screen the retrieved subproblems, while a single "master solver" is permitted to update $\Pi$. "Slave solvers" discard subproblems without improvements, while those yielding net gains are forwarded to the "master solver" for refining $\Pi$. The process continues iteratively until no improvement is observed in $K$ consecutive steps. Detailed pseudo-code is provided in Algorithm 2 in Appendix E.

## 4 EXPERIMENTS AND RESULTS ANALYSIS

### 4.1 EXPERIMENTAL SETUPS

**Evaluation datasets.** To comprehensively assess the performance of *ViTSP*, this work used TSP instances from TSPLIB (Reinelt, 1991) and a synthetic TSP-10K dataset with uniformly distributed nodes as primary evaluation datasets. The synthetic dataset contains 16 instances, following (Fang et al., 2024). TSPLIB offers a wide range of real-world instances for TSP, covering diverse distributions and scales. Moreover, Reinelt (2007) provides proven optimality for TSPLIB instances, enabling the measurement of optimality gaps even at very large scales. We chose TSP instances with $N \geq 1,000$ from the dataset to represent (very-)large-scale problems, where exact solvers begin to struggle. This results in 33 TSPLIB instances. These instances follow a naming format of `[keywords][number of nodes]`, such as `pla85900`. Instances with the same keywords are from the same application domain. Notably, since our framework does not require additional training or fine-tuning during implementation, we do not curate any training dataset.

None of the TSPLIB instances has been exposed to the baseline learning-based algorithms during their training phase. This ensures a fair evaluation of generalizability and scalability across all baselines.

To the best of our knowledge, our work provides one of the most comprehensive evaluations on this real-world benchmark dataset, offering a thorough assessment of the proposed *ViTSP*.

**Evaluation metrics**. We used two metrics to measure the performance of algorithms: (1) **Optimality gaps** (%); and (2) **Runtime** (in seconds). We used the reported proven optimal objective values $L^*$ (total distance traveled) for TSPLIB instances in Reinelt (2007) as reference and the gap is calculated as: $\frac{L_{\text{Model}} - L^*}{L^*} \times 100\%$, where $L_{\text{Model}}$ is the objective value produced by a baseline model. The recorded wall-clock runtime of *ViTSP* explicitly includes (1) the LKH initialization, (2) all VLM API waiting and latency, and (3) the Concorde solving time for subproblems. Thus, our reported time reflects the actual end-to-end wall time required by *ViTSP* in real-world settings.

***ViTSP* setups.** In the initialization module, we used LKH-3 with its default parameter settings to warm start the *ViTSP*. In the visual selection module, we employed GPT-4.1 (fast thinking VLM) and o4-mini (reasoning VLM) as the selectors in this study. We set the number of subproblems generated per prompt $Q = 2$. In the subproblem optimization module, Concorde, the SOTA exact solver, was utilized as the subproblem solver. *ViTSP* terminates when no improvement is observed in $K = 5$ consecutive steps, and the duration from initialization to termination is recorded as runtime. We ran *ViTSP* for five runs and obtained the average gaps and runtimes. For more detailed parameter configurations, please refer to Section F.5 in Appendix F.

**Baselines.** We compared our *ViTSP* against both classical OR approaches and learning-based approaches. Specifically, we applied the following ten baselines: (1) **Concorde**; (2) **LKH-3 (Default)**; (3) **LKH-3 (more `RUNS`)** (4) **FI**; (5) **AM**; (6) **DIFUSCO**; (7) **INViT**; (8) **SIT**; (9) **DeepACO**; (10) **SO**; (11) **UDC**; (12) **EoH**. Their description and implementation are provided in the Appendix F.

The selected learning-based approaches include both end-to-end solution construction methods and local improvement techniques that match the decomposition heuristics used in this study. They either provided open-source code and pre-trained checkpoints or reported results on TSPLIB instances (e.g., SO). The selected EoH produced applicable results on large-scale instances, whereas other LLM-based approaches, such as Yang et al. (2023); Elhenawy et al. (2024), failed to generate valid solutions even on small-scale cases and were therefore not included as baselines in this study.

We align the runtime across baselines to ensure fair comparison. In addition to using the default parameter values of LKH-3, we introduce LKH-3 (more `RUNS`), where the `RUNS` value is increased to match LKH-3's runtime with that of *ViTSP* on each instance. Similarly, Concorde, DeepACO, UDC, and EoH are run with the same or slightly longer time limit as *ViTSP*. For FI, runtime is deterministic with respect to instance size. End-to-end methods (AM, DIFUSCO, and INViT) also have deterministic runtimes, as they perform only a single feedforward inference. When running synthetic TSP-10K, we set the timelimit as 600 s to compare baselines' performances.

**Hardware.** We used an AMD EPYC 7443 24-Core CPU and an Nvidia L40 GPU with 48GB memory to implement our work and baseline algorithms. In *ViTSP*, the VLMs were accessed on demand online. Their usage did not rely on local GPU resources but was confined by the I/O rate.

## 4.2 MAIN RESULTS

We summarize the full performance comparison results in Table 1 (22 large TSPLIB instances) and Table 2 (11 very-large instances), reporting runtime (in seconds) and optimality gaps with the lowest gaps highlighted. The results of *ViTSP* are averaged over five runs. The selected box regions yielding gap reductions by VLMs are illustrated in Appendix H.

Overall, *ViTSP* achieves an average optimality gap of 0.24%, outperforming classical OR methods, including LKH-3 (more `RUNS`) at 0.31% and Concorde at 0.34%. In contrast to learning-based methods that fail to generalize to these unseen TSPLIB instances, *ViTSP* demonstrates consistently superior performance. More specifically, *ViTSP* attains the best performance on 20 (highlighted) out of 33 instances even when classical OR methods are allocated equal or longer runtimes. At instance-level, in relative to the optimality gaps of LKH-3, *ViTSP* further reduced gaps by 3.57% to 100.00%. While LKH-3 remains highly efficient for large instances with $1,000 < N < 4,000$, as the SOTA heuristic, the advantage of *ViTSP* becomes more pronounced as instance size further increases.

Since *ViTSP* is warm-started from LKH-3 (Default), we further compare the reduction of optimality gaps over time between *ViTSP* and LKH-3 (more `RUNS`) on selected instances of different scales in Figure 2. When given a small amount of additional runtime beyond LKH-3's default settings,

it reduces optimality gaps more rapidly than *ViTSP* in the early phase. However, its improvement quickly plateaus, whereas *ViTSP* continues to improve and eventually surpasses or matches LKH-3. As the problem size increases beyond 4,000 in Table 2, *ViTSP* significantly speeds up the reduction of optimality gaps and consistently reaches lower gaps compared to LKH-3. We present the complete results for optimality gap reduction over time across all TSPLIB instances between *ViTSP* and LKH-3 in Appendix G.

Table 1: Performance comparison on 22 large TSPLIB instances ($1000 \le n < 4000$).

| | | dsj1000 | pr1002 | u1060 | vm1084 | pcb1173 | d1291 | rl1304 | rl1323 | nrw1379 | fl1400 | u1432 |
|---|---|---|---|---|---|---|---|---|---|---|---|---|
| Concorde | Time(s) | 46.2 | 8.3 | 106.7 | 57.5 | 50.2 | 252.5 | 25.8 | 122.3 | 45.7 | 115.4 | 111.8 |
| | Gap | 0.00% | 0.00% | 0.00% | 0.00% | 0.00% | 0.65% | 0.00% | 0.03% | 0.00% | 0.24% | 0.03% |
| LKH-3 (Default) | Time(s) | 1.9 | 2.0 | 2.0 | 2.1 | 2.2 | 2.5 | 2.7 | 2.7 | 3.4 | 4.5 | 3.8 |
| | Gap | 0.17% | 0.47% | 0.53% | 0.12% | 1.04% | 0.61% | 0.55% | 0.21% | 0.61% | 0.19% | 0.54% |
| LKH-3 (RUNS) | Time(s) | 20.5 | 95.1 | 100.4 | 22.9 | 93.0 | 248.8 | 45.9 | 99.2 | 162.4 | 75.0 | 95.6 |
| | Gap | 0.00% | 0.00% | 0.01% | 0.00% | 0.01% | 0.00% | 0.00% | 0.05% | 0.01% | 0.18% | 0.03% |
| FI | Time(s) | 0.4 | 0.4 | 0.5 | 0.4 | 0.5 | 0.7 | 0.6 | 0.7 | 0.8 | 0.8 | 0.8 |
| | Gap | 11.23% | 10.30% | 12.45% | 9.51% | 15.27% | 21.50% | 22.99% | 20.78% | 11.29% | 4.17% | 12.71% |
| AM (G) | Time(s) | 0.8 | 0.7 | 0.8 | 0.8 | 0.9 | 1.0 | 1.0 | 1.1 | 1.1 | 1.1 | 1.1 |
| | Gap | 41.43% | 40.57% | 56.78% | 44.05% | 41.71% | 48.82% | 38.22% | 42.85% | 37.74% | 63.15% | 37.27% |
| DIFUSCO (S + 2-opt) | Time(s) | 23.6 | 23.1 | 23.9 | 25.8 | 29.3 | 31.9 | 33.1 | 34.7 | 36.1 | 33.1 | 37.4 |
| | Gap | 7.83% | 9.04% | 7.52% | 6.18% | 9.26% | 9.70% | 9.22% | 8.23% | 9.70% | 4.48% | 8.84% |
| INViT | Time(s) | 7.6 | 6.5 | 6.7 | 6.5 | 8.1 | 8.7 | 8.3 | 8.4 | 10.0 | 8.7 | 10.1 |
| | Gap | 8.65% | 10.55% | 9.74% | 6.61% | 6.85% | 8.92% | 8.97% | 8.33% | 6.57% | 13.86% | 5.09% |
| SIT | Time(s) | 70.1 | 66.0 | 57.2 | 38.9 | 124.3 | 169.2 | 27.4 | 95.7 | 104.8 | 54.1 | 83.5 |
| | Gap | 1.21% | 1.13% | 1.33% | 0.83% | 1.54% | 4.33% | 1.51% | 1.35% | 1.55% | 4.52% | 2.28% |
| DeepACO | Time(s) | 165.1 | 164.1 | 167.0 | 171.9 | 185.5 | 199.7 | 200.6 | 200.8 | 209.8 | 215.7 | 213.4 |
| | Gap | 21.51% | 20.96% | 37.97% | 34.59% | 20.40% | 24.85% | 35.00% | 29.60% | 18.92% | 45.59% | 14.75% |
| SO | Time(s) | N/A | N/A | 35.0 | 35.0 | 38.0 | N/A | 42.0 | 42.0 | 44.0 | 45.0 | N/A |
| | Gap | | | 2.21% | 2.20% | 2.87% | | 6.76% | 4.21% | 1.63% | 1.96% | |
| UDC | Time(s) | 47.0 | 91.5 | 65.2 | 53.2 | 140.6 | 242.8 | 42.1 | 123.2 | 148.6 | 70.2 | 89.2 |
| | Gap | 15.36% | 18.75% | 32.81% | 29.28% | 20.70% | 25.30% | 28.29% | 18.01% | 18.94% | 33.57% | 17.33% |
| EoH | Time(s) | 48.1 | 93.2 | 62.7 | 53.8 | 137.7 | 245.3 | 41.9 | 123.5 | 146.2 | 69.8 | 87.4 |
| | Gap | 448.08% | 56.86% | 596.53% | 6.47% | 104.29% | 18.31% | 89.73% | 50.14% | 94.52% | 26.38% | 35.39% |
| **ViTSP** | Time(s) | 69.6 | 65.0 | 57.1 | 38.5 | 124.3 | 168.6 | 27.2 | 95.6 | 104.6 | 53.9 | 83.2 |
| | Gap | 0.02% | 0.01% | 0.00% | 0.00% | 0.16% | 0.15% | 0.14% | 0.03% | 0.00% | 0.18% | 0.03% |

| | | fl1577 | d1655 | vm1748 | u1817 | rl1889 | d2103 | u2152 | u2319 | pr2392 | pcb3038 | fl3795 |
|---|---|---|---|---|---|---|---|---|---|---|---|---|
| Concorde | Time(s) | 262.7 | 24.4 | 189.4 | 199.8 | 112.3 | 100.0 | 314.7 | 443.9 | 13.1 | 390.6 | 600.0 |
| | Gap | 1.52% | 0.00% | 0.05% | 0.34% | 0.09% | 1.47% | 0.25% | 0.11% | 0.00% | 0.11% | 0.44% |
| LKH-3 (Default) | Time(s) | 4.0 | 4.5 | 5.3 | 5.1 | 6.1 | 6.1 | 7.4 | 10.7 | 11.4 | 15.3 | 25.1 |
| | Gap | 0.25% | 0.80% | 0.55% | 1.11% | 0.57% | 0.54% | 0.95% | 0.32% | 1.08% | 1.20% | 1.73% |
| LKH-3 (RUNS) | Time(s) | 16.3 | 150.0 | 180.0 | 202.3 | 118.8 | 104.5 | 331.5 | 316.4 | 282.1 | 407.6 | 527.4 |
| | Gap | 0.00% | 0.00% | 0.12% | 0.15% | 0.06% | 0.00% | 0.09% | 0.08% | 0.03% | 0.12% | 0.67% |
| FI | Time(s) | 1.0 | 1.2 | 1.2 | 1.3 | 1.4 | 1.7 | 1.8 | 2.1 | 2.4 | 4.0 | 5.6 |
| | Gap | 17.61% | 15.41% | 11.90% | 18.10% | 17.63% | 23.57% | 20.65% | 6.54% | 13.71% | 14.92% | 17.57% |
| AM (G) | Time(s) | 1.3 | 1.4 | 1.5 | 1.6 | 1.7 | 1.9 | 1.9 | 2.2 | 2.1 | 3.1 | 4.3 |
| | Gap | 51.92% | 61.15% | 49.56% | 56.51% | 49.57% | 55.48% | 66.28% | 29.92% | 62.35% | 62.33% | 84.55% |
| DIFUSCO (S + 2-opt) | Time(s) | 44.3 | 46.2 | 53.8 | 55.9 | 61.4 | 77.8 | 79.1 | 94.8 | 111.2 | 214.1 | 327.3 |
| | Gap | 7.66% | 10.33% | 8.93% | 13.47% | 8.53% | 12.65% | 13.94% | 5.72% | 10.74% | 10.95% | 6.98% |
| INViT | Time(s) | 10.7 | 12.7 | 11.8 | 13.4 | 13.1 | 15.9 | 17.1 | 18.8 | 20.7 | 30.0 | 36.4 |
| | Gap | 7.65% | 11.54% | 8.26% | 8.32% | 10.03% | 7.74% | 7.11% | 0.93% | 8.12% | 7.85% | 15.13% |
| SIT | Time(s) | 59.0 | 111.6 | 163.1 | 180.3 | 89.2 | 84.5 | 238.5 | 245.1 | 187.8 | 336.6 | 252.7 |
| | Gap | 5.04% | 5.36% | 1.17% | 2.28% | 4.01% | 5.60% | 4.21% | 0.22% | 1.12% | 1.95% | 7.55% |
| DeepACO | Time(s) | 236.8 | 241.7 | 250.8 | 262.9 | 276.3 | 296.3 | 299.0 | 321.1 | 333.6 | 418.0 | 506.7 |
| | Gap | 42.74% | 25.46% | 34.78% | 25.78% | 42.15% | 19.32% | 24.52% | 11.12% | 30.43% | 24.71% | 103.56% |
| SO | Time(s) | 51.0 | N/A | 53 | N/A | 58.0 | 71.0 | 66.0 | N/A | N/A | 93.0 | 121.0 |
| | Gap | 4.42% | | 3.61% | | 4.63% | 8.32% | 4.91% | | | 2.67% | 5.49% |
| UDC | Time(s) | 93.2 | 146.8 | 178.3 | 198.5 | 108.9 | 95.4 | 312.5 | 265.1 | 248.9 | 364.5 | OOM |
| | Gap | 24.94% | 19.55% | 33.24% | 24.43% | 36.28% | 15.79% | 27.32% | 21.52% | 28.25% | 26.98% | |
| EoH | Time(s) | 92.5 | 147.2 | 179.6 | 198.7 | 108.3 | 95.3 | 311.6 | 265.7 | 249.2 | 365.5 | 304.1 |
| | Gap | 181.75% | 26.82% | 3.47% | 18.07% | 515.05% | 279.06% | 57.30% | 50.31% | 11.71% | 33.82% | 59.46% |
| **ViTSP** | Time(s) | 58.9 | 111.2 | 162.8 | 180.0 | 89.0 | 84.0 | 238.3 | 244.9 | 187.6 | 336.3 | 252.4 |
| | Gap | 0.00% | 0.00% | 0.05% | 0.24% | 0.03% | 0.39% | 0.08% | 0.06% | 0.09% | 0.09% | 0.57% |

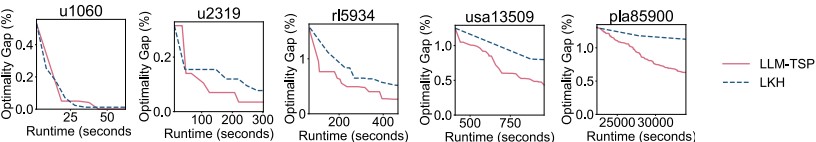

Figure 2: Optimality gaps over time on selected instances between ViTSP and LKH-3 (more RUNS).

Table 2: Performance comparison on 11 very-large TSPLIB instances ($4000 < n \leq 85900$).

| | | fnl4461 | rl5915 | rl5934 | pla7397 | rl11849 | usa13509 | brd14051 | d15112 | d18512 | pla33810 | pla85900 |
|---|---|---|---|---|---|---|---|---|---|---|---|---|
| Concorde | Time(s) | 305.0 | 550.0 | 266.0 | 671.3 | 1000.0 | 960.0 | 1742.0 | 2500.0 | 2941.0 | Failed | Failed |
| | Gap | 0.30% | 0.67% | 0.89% | 0.48% | 0.85% | 0.48% | 0.49% | 0.44% | 0.57% | | |
| LKH-3 | Time(s) | 39.8 | 63.3 | 64.0 | 92.0 | 311.8 | 382.9 | 417.5 | 524.3 | 728.2 | 2079.7 | 22344.0 |
| (Default) | Gap | 0.96% | 1.96% | 1.56% | 0.83% | 1.75% | 1.25% | 1.18% | 1.22% | 1.29% | 1.43% | 1.31% |
| LKH-3 | Time(s) | 292.9 | 549.3 | 548.1 | 673.5 | 1014.4 | 979.3 | 1908.5 | 2974.5 | 3176.2 | 8237.9 | 33966.5 |
| (RUNS) | Gap | 0.45% | 0.73% | 0.47% | 0.29% | 1.06% | 0.81% | 0.82% | 0.84% | 0.94% | 1.00% | 1.13% |
| FI | Time(s) | 8.85 | 14.96 | 15.42 | 22.26 | 60.50 | 82.51 | 89.38 | 104.11 | 155.37 | 462.16 | 3294.52 |
| | Gap | 11.30% | 22.15% | 20.91% | 13.24% | 19.39% | 12.52% | 11.64% | 11.67% | 11.77% | 16.84% | 14.46% |
| AM (G) | Time(s) | 5.18 | 7.97 | 8.23 | 11.06 | 23.51 | 29.63 | 31.62 | 35.46 | 50.77 | 153.60 | 959.18 |
| | Gap | 70.93% | 79.80% | 86.11% | 107.17% | 104.74% | 142.13% | 111.99% | 105.51% | 118.44% | 137.11% | 175.24% |
| DIFUSCO | Time(s) | 586.0 | 1321.0 | 1317.3 | 1946.9 | 5097.6 | 6598.8 | 7098.5 | 8226.8 | 12163.2 | OOM | OOM |
| (S + 2-opt) | Gap | 11.03% | 11.53% | 11.01% | 9.32% | 52.49% | 26.47% | 53.80% | 61.81% | 80.49% | | |
| INViT | Time(s) | 55.3 | 51.4 | 51.5 | 73.6 | 154.0 | 220.0 | 232.4 | 234.3 | 373.9 | 1010.4 | 5910.8 |
| | Gap | 6.58% | 9.43% | 10.84% | 7.66% | 10.19% | 11.94% | 9.21% | 8.04% | 8.38% | 7.34% | 6.34% |
| SIT | Time(s) | 277.4 | 386.5 | 485.7 | 575.6 | 785.2 | 995.5 | 1725.9 | 2173.2 | 2320.1 | 5759.3 | 29864.0 |
| | Gap | 1.42% | 3.46% | 5.09% | 1.96% | 5.01% | 12.34% | 4.39% | 3.30% | 3.73% | 4.40% | 6.60% |
| DeepACO | Time(s) | 594.7 | 766.9 | 763.7 | 1061.7 | 2513.4 | 3424.8 | 3818.8 | 4560.3 | 7665.8 | 33819.5 | OOM |
| | Gap | 37.31% | 73.03% | 77.79% | 76.26% | 94.37% | 130.03% | 102.88% | 84.05% | 100.42% | 160.79% | |
| SO | Time(s) | 139.00 | 194.00 | 196.00 | N/A | N/A | N/A | N/A | N/A | N/A | N/A | N/A |
| | Gap | 2.43% | 7.99% | 6.14% | | | | | | | | |
| UDC | Time(s) | 304.1 | OOM | OOM | OOM | OOM | OOM | OOM | OOM | OOM | OOM | OOM |
| | Gap | 12.80% | | | | | | | | | | |
| EoH | Time(s) | 305.1 | 539.2 | 468.2 | 739.2 | 725.6 | 961.0 | 1744.5 | 2501.7 | Failed | Failed | Failed |
| | Gap | 3.10% | 15.15% | 26.68% | 23.72% | 114.86% | 60.94% | 70.64% | 11.41% | | | |
| **ViTSP** | Time(s) | 277.3 | 386.5 | 485.5 | 575.5 | 785.1 | 995.2 | 1725.3 | 2172.0 | 2319.0 | 5758.1 | 29863.6 |
| | Gap | 0.16% | 1.13% | 0.41% | 0.28% | 1.03% | 0.47% | 0.31% | 0.22% | 0.36% | 0.52% | 0.83% |

Learning-based methods struggle to generalize their learned policies to OOD instances, leading to inferior performance compared to *ViTSP*. Large gaps remain across the baselines, which impair their utilization at a practical scale. For example, AM and DeepACO underperform the simple heuristic algorithm FI, demonstrating their brittleness when no model reconfiguration or time-consuming retraining is performed. Furthermore, due to their high GPU memory requirements, all learning-based algorithms except AM and INViT fail to scale to pla85900 and encounter out-of-memory (OOM) errors. Notably, UDC suffers from OOM when the instance size exceeds 5,000, failing to produce feasible decomposition heuristics for very large-scale TSPs. For LLM-based approaches, EoH shows limited effectiveness in designing heuristics for varying TSP instances, exhibiting high variance in optimality gaps across all TSPLIB instances in both Table 1 and Table 2. These observations highlight the advantage of *ViTSP*, which leverages generative models to visually guide high-quality decomposition heuristics while reducing reliance on local GPU resources and (re-)training.

The results on very-large-scale synthetic TSP (Table 3) further confirm the effectiveness of our proposed *ViTSP*, matching with the superior performance of *ViTSP* in TSPLIB instances with n>10,000. Notably, *ViTSP* outperforms standalone LKH-3 and other learning-based methods. Even though SIT has been trained on TSP-10K, it is still less effective than *ViTSP* at such a scale even longer runtime budgets were given.

Table 3: Performance comparison on uniform instances with $n = 10,000$.

| | Near-optimality | LKH-3 (more RUNs) | FI | AM(G) | DIFUSCO (S + 2-opt) | INViT | SIT (PRC,1000) | DeepACO | SO | UDC | ViTSP |
|---|---|---|---|---|---|---|---|---|---|---|---|
| Obj. | 71.78 | 72.54 | 80.59 | 141.68 | 73.89 | 76.09 | 73.08 | 79.76 | N/A | OOM | 72.28 |
| Gap | — | 1.06% | 12.27% | 97.38% | 2.94% | 6.01% | 1.81% | 11.12% | — | — | 0.70% |
| Time (s) | — | 645 | 55.3 | 17.5 | 610 | 30.9 | 1020 | 605.1 | — | — | 615.7 |

Our experiments suggest that certain TSP structures can make optimization more or less difficult. Although pr2392 is twice the size of prc1173, Concorde uses only 26% of the time required for prc1173 to reach optimality for pr2392. Also, both *ViTSP* and LKH-3 struggle to find high-quality solutions on fl1400, and *ViTSP* shows difficulty in reducing gaps on d2103 and rl5915.

To further elucidate the efficiency and practical cost of *ViTSP*, we provide a detailed analysis of iteration-level behavior, valid subproblem rates, and VLM API expenses in Appendix I.

## 4.3 ABLATION STUDIES

**The effectiveness of VLMs.** To verify that VLMs can conduct principled selection as visual selectors, rather than fruitless permutation, we devise two heuristic selection policies adopted from Li et al. (2021) and Cheng et al. (2023): (1) **Random sequence selector**: uniformly and randomly selecting a segment of a given length from the tour. The segment length is set to match the average number

of nodes selected per step by *ViTSP*. (2) **Random box selector**: uniformly and randomly selecting rectangular subproblems of random sizes. It chooses $Q = 2$ subproblems at each step, the same as in *ViTSP*. We replace the default VLM selector modules with these alternatives in *ViTSP* and execute the same experiments on a given instance, where all scenarios start with solutions initialized using LKH-3. Due to the page limit, we report the optimality gaps over time using different selector policies on selected instances in Figure 3. The full results are illustrated in Appendix J.

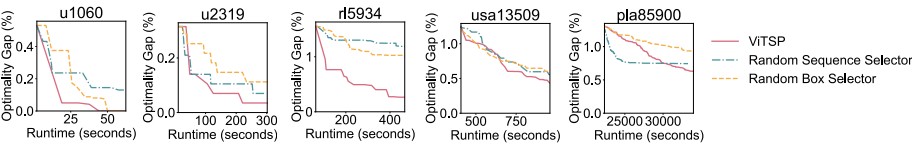

Figure 3: Ablation studies of different selection policies on selected instances.

VLMs are capable of performing meaningful, non-random subproblem selection as prompted in the framework *ViTSP*. As shown in Figure 3, *ViTSP* consistently reduces optimality gaps over time and outpaces both random sequence and random box methods. This highlights the effectiveness of visually leveraging instance-specific structures to guide subproblem selection and generalize to (very-) large-scale unseen TSP instances.

While the two random selection strategies demonstrate some ability to reduce gaps, they consistently converge to local optima. Interestingly, for `pla85900`, the random sequence selection outperforms *ViTSP* in the early stages by reducing the gap more rapidly. This suggests the potential value of alternative operations beyond the box-region selection used in this study.

**The effect of solution initialization.** We examine the impact of solution initialization on the overall performance of *ViTSP*. Based on the same runtime budget, we run *ViTSP* using LKH-3 with default parameters and FI, respectively, to initialize the solution. We report the average optimality gaps based on five runs in Table 4. The results show that the ViTSP initialized by LKH consistently outperforms the one initialized by FI, since LKH is known for providing higher quality solutions. These results confirm that while initialization affects performance, the VLM-guided decomposition consistently improves over its corresponding initialization baseline.

Table 4: Optimality gap comparison of *ViTSP* under LKH and FI initializations

|  | dsj1000 | pr1002 | u1060 | vm1084 | pcb1173 | d1291 | rl1304 | rl1323 | nrw1379 | fl1400 | u1432 |
|---|---|---|---|---|---|---|---|---|---|---|---|
| Time (s) | 69.6 | 65.0 | 57.1 | 38.5 | 124.3 | 168.6 | 27.2 | 95.6 | 104.6 | 53.9 | 83.2 |
| LKH | 0.02% | 0.01% | 0.00% | 0.00% | 0.16% | 0.15% | 0.14% | 0.04% | 0.00% | 0.18% | 0.03% |
| FI | 0.50% | 0.03% | 0.64% | 0.67% | 0.21% | 2.42% | 8.01% | 0.39% | 0.01% | 1.30% | 1.43% |

|  | fl1577 | d1655 | vm1748 | u1817 | rl1889 | d2103 | u2152 | u2319 | pr2392 | pcb3038 | fl3795 |
|---|---|---|---|---|---|---|---|---|---|---|---|
| Time (s) | 58.9 | 111.2 | 162.8 | 180.0 | 89.0 | 84.0 | 238.3 | 244.9 | 187.6 | 336.3 | 252.4 |
| LKH | 0.00% | 0.00% | 0.05% | 0.24% | 0.03% | 0.39% | 0.08% | 0.06% | 0.09% | 0.09% | 0.57% |
| FI | 17.61% | 6.56% | 0.04% | 3.16% | 0.50% | 23.55% | 0.98% | 0.07% | 0.41% | 0.34% | 3.02% |

|  | fnl4461 | rl5915 | rl5934 | pla7397 | rl11849 | usa13509 | brd14051 | d15112 | d18512 | pla33810 | pla85900 |
|---|---|---|---|---|---|---|---|---|---|---|---|
| Time (s) | 277.3 | 386.5 | 485.5 | 575.5 | 785.1 | 995.2 | 1725.3 | 2172.0 | 2319.0 | 5758.1 | 29863.6 |
| LKH | 0.16% | 1.13% | 0.41% | 0.28% | 1.03% | 0.47% | 0.31% | 0.22% | 0.36% | 0.52% | 0.83% |
| FI | 2.49% | 8.42% | 6.40% | 5.72% | 9.36% | 8.89% | 0.52% | 0.24% | 0.54% | 10.65% | 2.81% |

## 5 CONCLUSIONS

In this study, we proposed a vision-guided framework to effectively solve TSPs with varying scales and distributions. *ViTSP* hybridizes the strength of pre-trained VLMs and existing OR techniques by selecting instance-specific subproblems visually and then delegating them to an off-the-shelf solver. Our proposed *ViTSP* bypasses *ad hoc* training while exhibiting effectiveness and scalability, achieving lower average optimality gaps than LKH-3 and baseline learning-based approaches. Because TSP serves as the base case for a wide family of routing problems, the promising results from ViTSP suggest opportunities to expand our framework to other routing problems. While this study validates the effectiveness of visual guidance, interpreting *how* the decomposition is determined lies beyond our current scope and remains an important future direction. Additionally, lightweight fine-tuning of offline VLMs, and further exploration of in-context reinforcement learning, represent promising future directions to further enhance performance of VLMs for TSP and broad routing problems. Further limitations are discussed in the Appendix K. More broadly, this work highlights the potential of using generative AI to support CO at practical scales, particularly in settings where abundant training data is unavailable.

## ETHICS STATEMENT

I acknowledge that I and all co-authors of this work have read and commit to adhering to the ICLR Code of Ethics. To the best of our knowledge, this work does not involve potential violations of the ICLR Code of Ethics.

## REPRODUCIBILITY STATEMENT

We have made efforts to ensure the reproducibility of this work. The source code is available at https://github.itap.purdue.edu/uSMART/ViTSP_ICLR2026. Details of the experimental setup are provided in Section 4.1 of the main text and in Sections C and F.5 of the Appendix to further support reproducibility. Additionally, the TSPLIB dataset used in this study is publicly available, which ensures the reproducibility of the results.

## THE USE OF LARGE LANGUAGE MODELS (LLMS)

The authors confirm that LLMs were used only for grammar checking and text polishing. They were not involved in research ideation. Their role in writing was limited, such that they are not considered contributors.

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

## A    TRAVELING SALESMAN PROBLEM (TSP)

A TSP is characterized by a list of nodes $i \in \{1, 2, ..., N\}$, and the corresponding coordinate sets $\{(x_i, y_i) \mid i = 1, 2, \ldots, N\}$ or the 2D Euclidean distance matrix $D_N$. The 2D Euclidean distance between a node pair is calculated as $d(i,j) = [\sqrt{(x_j - x_i)^2 + (y_j - y_i)^2}]$ and we round the distance to the nearest integer in this study.

The goal of TSP is to find an optimal tour that departs from an initial node, visits each node exactly once, and returns to the starting node. Without loss of generality, the solution $\Pi$ in this study is represented as a cyclic sequence of nodes $\Pi = [\pi_1, \pi_2, ..., \pi_N] + [\pi_1]$, where $\pi_n$ represents the *n-th* node in this sequence, and $\pi_1$ denotes both the starting and ending node of the tour to form a complete cycle. The objective is to minimize the total distance traveled $L(\Pi) = \sum_{i=1}^{N-1} d(\pi_i, \pi_{i+1}) + d(\pi_N, \pi_1)$.

When the distance between two nodes is identical in both directions, i.e., $d(i,j) = d(j,i)$, the problem is known as symmetric TSP (STSP). In contrast, asymmetric TSP (ATSP) allows for different distances between certain node pairs in opposite directions, i.e., $d(i,j) \neq d(j,i)$.

## B    ADDITIONAL RELATED WORKS

### B.1    OR APPROACHES

Many commercial solvers, such as Gurobi, OR-Tools, and CPLEX, are designed for generic optimization purposes. They search for exact optimal solutions using techniques like branch and cut, but these solvers struggle with large-scale optimization problems. Besides these commercial solvers, Concorde is believed to be the SOTA exact solver designed for TSP to obtain optimal solutions. Essentially, it also employs the LKH algorithm–the SOTA heuristic solver–and branch-and-cut techniques to find exact solutions. Concorde has been shown to solve TSPs with more than 80k nodes, but still at the expense of years of computation.

### B.2    LEARNING-BASED APPROACHES

**End-to-end construction.**   The autoregressive approach is characterized by an attention-based network architecture, such as the Transformer (Vaswani et al., 2017), or its variants, such as Pointer-former (Jin et al., 2023). AM in Kool et al. (2019) uses the REINFORCE algorithm to sequentially predict the node with the highest probability, while POMO in Kwon et al. (2020) produces multiple solutions in decoding steps to improve the model performance. In contrast, the non-autoregressive approaches estimate the likelihood of connecting each edge between nodes to produce a heatmap. For example, DIMES in Qiu et al. (2022) proposed learning a continuous space to parameterize the solution distribution using an anisotropic graph neural network. DIFUSCO (Sun & Yang, 2023), T2T (Li et al., 2023), and Fast T2T (Li et al., 2024) developed a graph-based diffusion model to generate the solution, which denoises random noise and the problem instance to gradually produce a feasible solution. However, DIFUSCO employs the 2-opt method to further refine the output, which brings a significant performance gain. Its standalone performance to generalize to new instances is questionable. Fang et al. (2024) introduced Invariant Nested View Transformer (INViT) to identify partial nodes with similar distributions as the trained ones to hierarchically handle partial problems. However, since these solvers always generate approximate solutions with an inevitable optimality gap, the overall solution quality can deteriorate significantly in out-of-distribution instances.

**Local improvement.**   Li et al. (2021) trains a backbone to select a promising subproblem and then delegates it to an off-the-shelf solver for further improvement. Zong et al. (2022) developed Rewriting-by-Generating (RBG) to iteratively refine the solution partitioning and infer new local solutions by a trained generator. Similarly, Select-and-Optimize (SO) in Cheng et al. (2023) trained a policy to select promising sequences within the complete tour and used a trained solver to improve the selected sequence iteratively. Intuitively, the subproblem solution quality fully depends on the solver. RBG and SO trained small-scale solvers following the methods in end-to-end construction. Fu et al. (2021) developed a graph convolutional residual neural network with attention mechanisms (AttGCRN) to optimize split subgraphs, and it fuses optimized partial solutions as the complete solution. Similarly, H-TSP (Pan et al., 2023), GLOP (Ye et al., 2024b), and UDC (Zheng et al., 2024)

decompose a TSP into open TSPs (instead of standard symmetric TSP) and optimize them using dedicated trained solvers.

## C    PROMPTS TO VLMS FOR SUBPROBLEM VISUAL SELECTION

The meta-text prompt instructing VLMs to select promising TSP subproblems is devised as follows. ***Italicized text in bold*** denotes placeholders for problem-specific inputs, and $Q$ represents the number of sub-regions to be selected per query. $\Delta$ indicates the margin used to allow selection near the edge of the instance. It is set to 10% of the spatial boundary of the given TSP instance. We do not set parameters or rules for exploring, to investigate VLMs' capability to learn from the context to determine the best selection by itself. Due to the inherent differing execution pace between modules in asynchronous orchestration, VLMs can generate selections on the same global solutions before solvers finish current subproblem jobs. To mitigate duplicate selections from VLMs during asynchronous processes, the real-time subproblem queue $\Omega$ is included as part of the input prompts to VLMs.

> You are tasked with improving an existing solution to a Traveling Salesman Problem (TSP) by selecting a sub-region where the routes can be significantly optimized. Carefully consider the locations of the nodes (in red) and connected routes (in black) in the initial solution on a map. The boundary of the map is x_min= ***{x_min - $\Delta$}***, x_max= ***{x_max + $\Delta$}***, y_min= ***{y_min - $\Delta$}***, y_max= ***{y_max + $\Delta$}***.
>
> Please return ***{Q}*** sub-rectangle(s) that you believe would most reduce total travel distance from further optimization by a downstream TSP solver. Analyze the problem to do meaningful selections. Remember, if you don't see significant improvement, try selecting larger areas that cover more nodes based on your analysis of the prior selection trajectory.
> Keep your output very brief as in the following template. Don't tell me you cannot view or analyze the map. I don't want an excuse:
> $\langle$coordinates$\rangle$ x_min=1,000 , x_max=2,000 , y_min=1,000 , y_max=2,000 $\langle$/coordinates$\rangle$
>
> Avoid selecting the same regions as follows, which are pending optimization: ***{pending regions}***
> Below are some previous selection trajectories. Please avoid selecting the same subrectangle: ***{prior selection trajectory}***

where each entry in ***{pending regions}*** is retrieved from up-to-date subproblem queue $\Omega$. Each entry in the ***prior selection trajectory*** is based on the trajectory pool $\Phi$ and formatted as follows:

> ***{coordinates}***, number of nodes within the subrectangle= ***{number of nodes}***, travel distance reduction= ***{delta objective improvement}***, computational time for this subproblem= ***{solver runtime in second}***

The visual prompt is an instance-specific image, visualizing the position of nodes and the current tour based on $\Pi^*$. An example visual prompt is shown in Figure 4. In our implmentation, all TSP instances are rendered using a consistent figure size (figsize=(20,20)). The image is gridded based on $\left\lceil \sqrt{\frac{y_{max} - y_{min}}{100}} \right\rceil$ to adaptively provide coordinate reference for VLM.

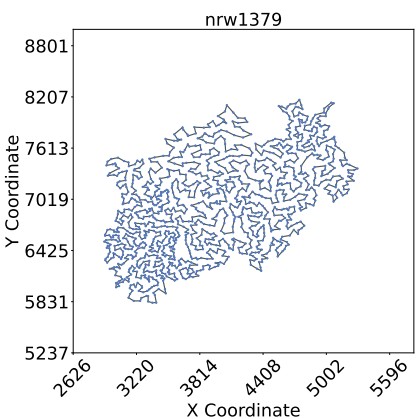

Figure 4: An example of the visual prompt to VLMs. In this example, `nrw1379` is used. The tour is initialized by LKH-3.

## D    PSEUDO-CODES FOR VISUAL SELECTION MODULE

---

**Algorithm 1** Visual Selection Module

---

1: **Input:** Current global solution $\Pi^*$, Selection trajectory $\Phi$, Meta-instruction $I$, Pending subproblem queue $\Omega$, Number of subproblems per prompt $Q$, Maximum number of covered nodes $\alpha$, VLM visual selection $F_{\text{selector}}$.
2: $(C_1, \ldots, C_Q) \leftarrow F_{\text{selector}}(\Pi^*, \Phi, I, \Omega, Q)$
3: **for** $q = 1$ **to** $Q$ **do**
4:     Compute the number of covered nodes $M$
5:     **while** $M > \alpha$ **do**
6:         $C_q \leftarrow F_{\text{selector}}(\Pi^*(C_q), \Phi, I, \Omega, 1)$       ▷ Zoom-in based on current solution within $C_q$
7:         Update $M$
8:     **end while**
9:     $\omega_q = (W_q, K_q) \leftarrow \text{FORMSUBPROBLEMS}(\Pi^*, C_q)$
10: **end for**
11: **if** $\left| \bigcup_{q=1}^{Q} W_q \right| \le \alpha$ **or** $\exists i \ne j$ such that $W_i \cap W_j \ne \emptyset$ **then**     ▷ Too small or overlapping subproblems
12:     $\omega \leftarrow \omega_1 \cup \cdots \cup \omega_Q$                              ▷ Merge subproblems
13:     Enqueue $\omega$ into $\Omega$
14: **else**
15:     **for** $q = 1$ **to** $Q$ **do**
16:         Enqueue $\omega_q$ into $\Omega$
17:     **end for**
18: **end if**
19: Return $\Omega$                                       ▷ Updated subproblem queue

---

# E PSEUDO-CODES FOR ASYNCHRONOUS ORCHESTRATION

---

**Algorithm 2** Asynchronous Orchestration of *ViTSP*

---

1: **Input:** Distance matrix $D_N$, meta-instruction $I$, subproblems per prompt $Q$, node cap $\alpha$, initializer $F_{\text{initializer}}$, VLM family $(\text{VLM}_1, \text{VLM}_2, \dots)$, solver $F_{\text{solver}}$, max no-improvement steps $K$, number of parallel slave solvers $P$.
2: $\Pi^* \leftarrow F_{\text{initializer}}(D_N)$                                                                ▷ Sec. 3.2
3: $\Phi \leftarrow \emptyset, \Omega \leftarrow \emptyset, \Omega' \leftarrow \emptyset$ ▷ Selection trajectory, subproblem queue, screened subproblem queue
4: **[Parallel: Visual Selection Loop]**                                                       ▷ Sec. 3.3
5: $F_{\text{selector}}(\cdot) \leftarrow \text{ROULETTE}(\text{VLM}_1, \text{VLM}_2, \dots)$
6: $\Omega \leftarrow \text{VISUALSELECTION}(\Pi^*, \Phi, I, \Omega, Q, \alpha, F_{\text{selector}})$                       ▷ Alg. 1
7: **[Parallel: $P$ Slave Solvers Loop]**                                                     ▷ Sec. 3.5
8: Dequeue $\omega$ from $\Omega$
9: $D_{STSP} \leftarrow \text{REFORMULATESUBPROBLEMS}(\omega)$
10: $\Pi \leftarrow F_{\text{solver}}(D_{STSP})$
11: **if** $L(\Pi^*) > L(\Pi)$ **then** Enqueue $\omega$ into $\Omega'$           ▷ Retain promising subproblem
12: **[Parallel: One Master Solver Loop]**                                                    ▷ Sec. 3.5
13: **while** Counter $\leq K$ **do**
14:     **if** $\Omega' \neq \emptyset$ **then** Dequeue $\omega$ from $\Omega'$
15:     **else** Dequeue $\omega$ from $\Omega$
16:     $D_{STSP} \leftarrow \text{REFORMULATESUBPROBLEMS}(\omega)$
17:     $\Pi \leftarrow F_{\text{solver}}(D_{STSP})$
18:     **if** $L(\Pi^*) > L(\Pi)$ **then** $\Pi^* \leftarrow \Pi$           ▷ Only master updates global solution
19:     **else** Counter $\leftarrow$ Counter $+ 1$
20: **end while**
21: **Terminate all parallel processes**                                                      ▷ Stop *ViTSP*
22: Return $\Pi^*$                                                                             ▷ Final solution

---

# F    IMPLEMENTATION DETAILS OF VITSP AND BASELINE ALGORITHMS

## F.1    CLASSICAL OR APPROACHES

**Concorde** We used *PyConcorde*, a Python wrapper for the Concorde solver, to solve the TSP instances. Concorde provides a `timelimit` parameter that can be used to constrain its execution time. The runtime limit for Concorde is set to match the time *ViTSP* takes to converge on each specific instance.

**LKH-3** The implementation version used in this study is *LKH-3.0.13*. For **LKH-3 (Default)**, we used the default parameters as specified in Helsgaun (2016), including RUNS=10, MAX_TRIALS=number of nodes, and MOVE_TYPE=5 (i.e., 5-opt). For **LKH-3 (more RUNS)**, we incrementally increase the value of RUNS by 50 until the actual runtime matches that of *ViTSP*. In the event that the gap is 0%, we report the runtime to reach this optimality.

LKH-3's performance is controlled by parameters like RUNS, TRIALS, etc. By default, RUNS is set to 10, which is used for LKH-3 (default) in our study. TRIALS is always set to be equivalent to the problem size, e.g., TRIALS=1000 for dsj1000. We extend RUNS ad hoc to ensure LKH-3 runtime matches with *ViTSP*, and we report the per-instance settings of RUNS in Table 5.

Table 5: Per-instance setting of RUNS across 33 TSPLIB instances.

| dsj1000 | pr1002 | u1060 | vm1084 | pcb1173 | d1291 | rl1304 | rl1323 | nrw1379 | fl1400 | u1432 |
|---------|--------|-------|--------|---------|-------|--------|--------|---------|--------|-------|
| 160     | 910    | 750   | 160    | 660     | 1600  | 260    | 360    | 760     | 210    | 410   |

| fl1577 | d1655 | vm1748 | u1817 | rl1889 | d2103 | u2152 | u2319 | pr2392 | pcb3038 | fl3795 |
|--------|-------|--------|-------|--------|-------|-------|-------|--------|---------|--------|
| 60     | 560   | 560    | 650   | 310    | 310   | 760   | 460   | 460    | 460     | 310    |

| fnl4461 | rl5915 | rl5934 | pla7397 | rl11849 | usa13509 | brd14051 | d15112 | d18512 | pla33810 | pla85900 |
|---------|--------|--------|---------|---------|----------|----------|--------|--------|----------|----------|
| 110     | 130    | 130    | 130     | 44      | 30       | 50       | 70     | 50     | 50       | 18       |

**Farthest insertion (FI)** No parameters required for this algorithm.

## F.2    END-TO-END CONSTRUCTION APPROACHES

**Attention Model (AM)** The algorithm was implemented based on RL4CO package (Berto et al., 2024). We utilized the open-source *tsp100* checkpoint as the backbone for the AM (Kool et al., 2019). Attempts to train a new model for larger instances, like *tsp1000*, failed due to an out-of-memory issue. We adopted the *greedy* decoding strategy since our experiments show it demonstrated superior performance compared to the *sampling* strategy. We denoted this algorithm as *AM(G)*.

**DIFUSCO** (Sun & Yang, 2023) We used the published pre-trained checkpoint *tsp10000-categorical* as the backbone for DIFUSCO in this study. All other parameters followed the defaults provided in the open-source code. Specifically, a sampling-based strategy was implemented with 10 diffusion steps and 16 samples. Additionally, 2-opt operations with 5,000 steps were applied. The diffusion type was categorical. We refer to this algorithm as *DIFUSCO(S+2-opt)*.

**Invariant Nested View Transformer (INViT)** (Fang et al., 2024) We used the open-sourced checkpoint for TSP as the backbone in assessing INViT's performance in this study and followed the default settings.

**Self-Improved Training (SIT)** (Luo et al., 2025) We used the open-sourced checkpoint *tsp-1k* as the backbone. Random Destroy and Repair (PRC) iterations were performed to improve the solutions until the runtime hits the limit aligned with *ViTSP* for each instance.

## F.3    LOCAL IMPROVEMENT APPROACHES

**DeepACO** (Ye et al., 2023) We used an open-sourced checkpoint *tsp500*. Following the specifications in the paper, we use the configuration $n_{ants} = 100$, $n_{nodes} = 500$, $k_{sparse} = 100$, $t_{aco} = 100$.

**Select-and-Optimize (SO)** (Cheng et al., 2023) No public codes and checkpoints are available. We extracted the results reported in the original paper.

**Unified Neural Divide-and-Conquer Framework (UDC)** (Zheng et al., 2024) We used their pre-trained checkpoints for partitioning (dividing) and sub-TSP solver (conquering) to evaluate UDC's

performance in TSP. We followed the default parameters reported in the paper. We set the runtime to align with *ViTSP*'s runtime on each instance.

### F.4 LLM-BASED APPROACH

**EoH** (Liu et al., 2024) We used the open-source `LLM4AD` platform to run EoH with its default parameters. For LLM, we used GPT-4.1, the same as in *ViTSP*. Additionally, iterations are terminated once the runtime limit, matched to that of *ViTSP*, is reached.

### F.5 VITSP

We used LKH-3 (Default) as the solution initializer. For VLMs, we set the number of subproblems generated per prompt $Q = 2$. To reconcile the solving time spent on a single subproblem, we set the upper bound of the number of selected nodes to $\alpha = 1000$ for instances $N < 10,000$ and $\alpha = 2000$ for instances $N > 10,000$ and we imposed the time limit on Concorde for solving each subproblem to be $T_{\max} = 10$ seconds. The number of slave solvers is set to be $P = 8$ in the asynchronous orchestration. For the VLM, a maximum of 100 tokens is set to ensure brief and speedy output. *ViTSP* is set to terminate if there are five consecutive solving steps without any improvement in the global objective value.

# G COMPLETE PLOTS OF OPTIMALITY GAP REDUCTION OVER TIME BETWEEN VITSP AND LKH-3 (MORE RUNS)

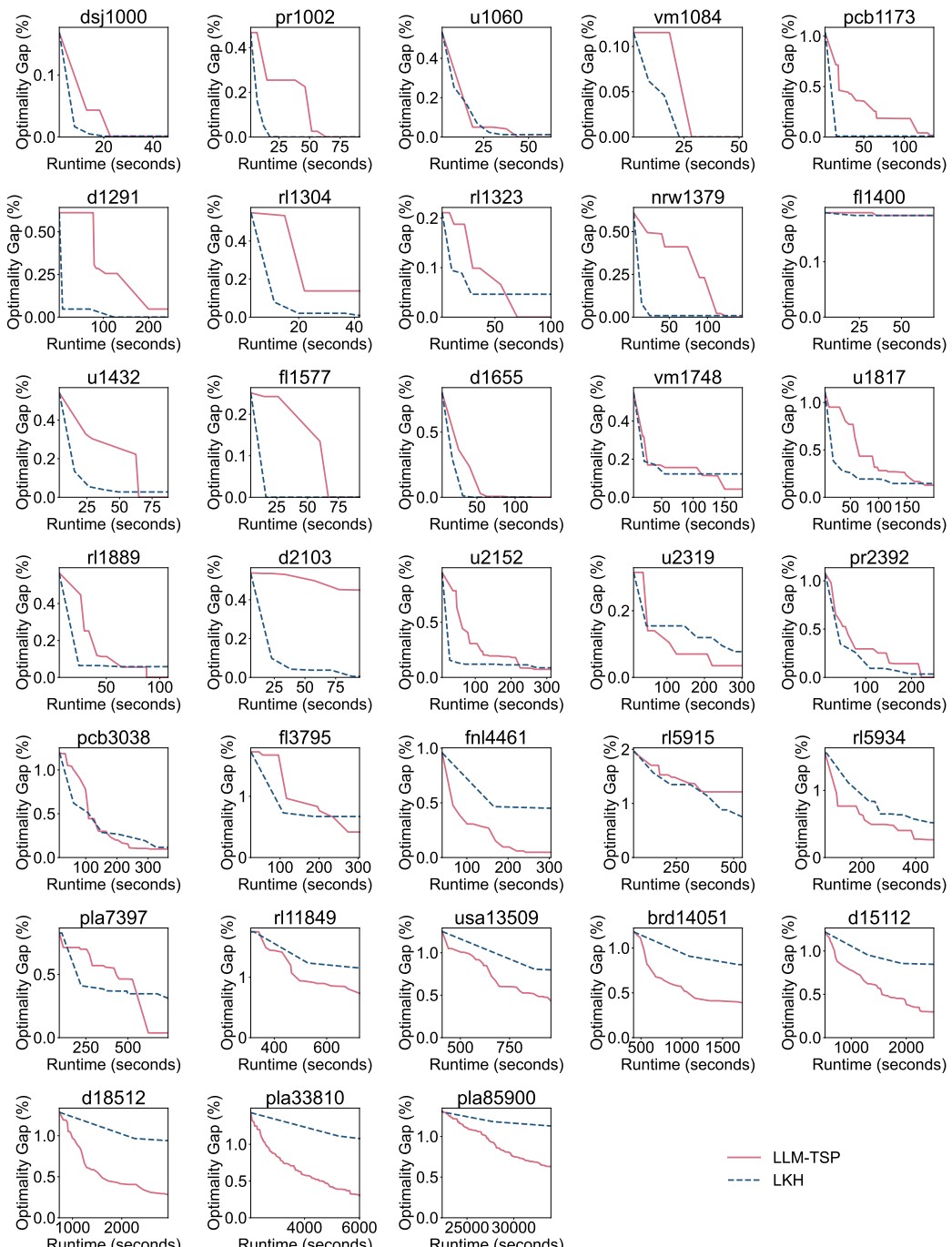

Figure 5: Optimality gap reduction over time between *ViTSP* and LKH-3 (more RUNS).

## H    IDENTIFIED SUBPROBLEMS BY VLMS THAT CONTRIBUTE TO OPTIMALITY GAP REDUCTION

In Figure 6, we plot the identified subproblems by VLM. For clarity, the selected box regions without contributing to optimality gap reductions are omitted.

Besides correcting crisscrossed edges, which is the most visually apparent suboptimality, *ViTSP* can also achieve improvements through:

(1) Refining dense regions where small gains can accumulate by transitioning from a locally approximate solution (initialized by LKH-3) to a locally optimal one, as each subproblem is re-optimized using an exact solver (Concorde).

(2) Combinatorially selecting neighborhoods whose joint optimization can escape the local optimum and lead to global improvements, recall that we use $Q = 2$ to allow two subproblem selections simultaneously at each step.

As shown in Figure 6, the subregions identified by the VLM often correspond to these two patterns. Dense areas may contain sub-paths that are locally optimal but globally improvable when viewed in context. Additionally, as we detailed in Section 3.3, we designed zoom-in reselection so VLMs always have chances to inspect detailed edge connections in a dense area.

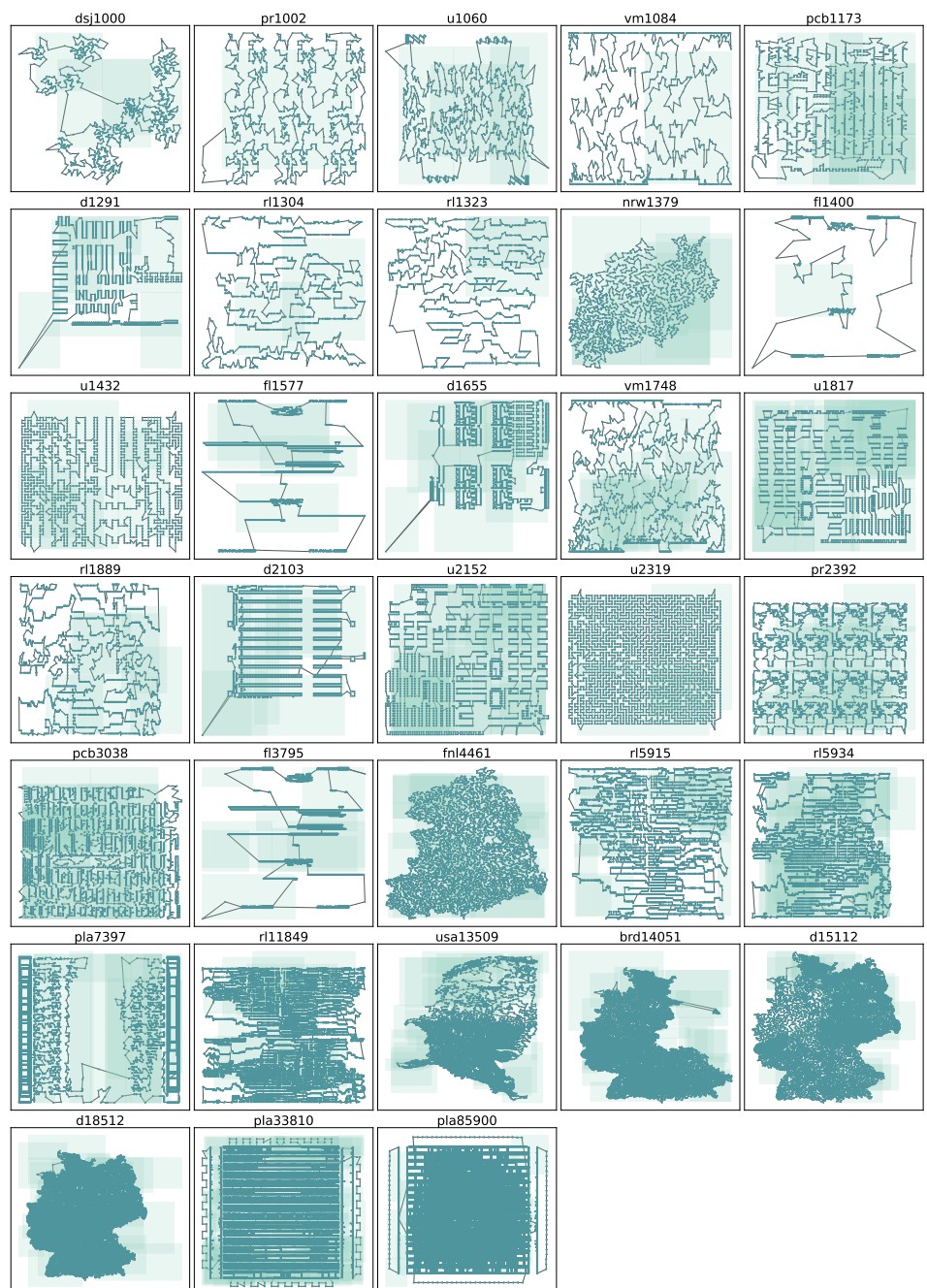

Figure 6: Visualization of selected box regions by VLMs on TSP instances. Darker shaded areas represent more frequently selected subproblems

# I   ANALYSIS OF *ViTSP* ITERATIONS

**Valid subproblem rate.** We here report the average valid subproblem rate per instance by VLMs over five runs of *ViTSP*. The valid rate is the ratio of subproblems that lead to gap reductions after optimization to the total number of subproblems proposed by the VLMs during the iterations. As shown in Table 6, the average valid rate ranges between 10% to 50%. These valid rates are achieved without any task-specific training, as the VLMs used in this study are entirely general-purpose. Overall, our results demonstrate that VLMs can generate practically meaningful decomposition

heuristics. Parameters such as box size, number of boxes per call, and subregion shape offer additional room for optimization.

The visual and textual prompts in *ViTSP* play complementary and essential roles in eliciting effective subproblems selection from VLMs. Prior works have shown that relying solely on textual inputs, such as approaches use only coordinate lists, such as in Yang et al. (2023) and Liu et al. (2024), struggle to produce high-quality solutions. Conversely, multimodal approaches like Elhenawy et al. (2024) attempt to read specific node indices from TSP images through VLMs are unable to scale effectively as visual clutter and index ambiguity grow rapidly with problem size. Together, these observations highlight the necessity of carefully designed multimodal prompting to achieve valid and effective performance throughout the *ViTSP* iterations.

Table 6: Valid rate of subproblems selected by VLMs across 33 TSPLIB instances.

| dsj1000 | pr1002 | u1060 | vm1084 | pcb1173 | d1291 | rl1304 | rl1323 | nrw1379 | fl1400 | u1432 |
|---|---|---|---|---|---|---|---|---|---|---|
| 41.72% | 53.81% | 48.14% | 35.93% | 35.60% | 10.42% | 41.67% | 30.15% | 39.40% | 33.00% | 34.92% |

| fl1577 | d1655 | vm1748 | u1817 | rl1889 | d2103 | u2152 | u2319 | pr2392 | pcb3038 | fl3795 |
|---|---|---|---|---|---|---|---|---|---|---|
| 48.91% | 25.44% | 34.26% | 24.75% | 31.14% | 50.00% | 22.64% | 12.83% | 37.47% | 30.66% | 20.50% |

| fnl4461 | rl5915 | rl5934 | pla7397 | rl11849 | usa13509 | brd14051 | d15112 | d18512 | pla33810 | pla85900 |
|---|---|---|---|---|---|---|---|---|---|---|
| 43.99% | 23.06% | 37.62% | 29.05% | 17.02% | 25.47% | 36.84% | 39.22% | 35.89% | 38.90% | 40.37% |

**Cost of VLM API calls per instance.** In *ViTSP*, the primary cost comes from online VLM API calls. The cost of an API call depends on the input and output tokens. For GPT-4.1, the pricing is $2 for every one million input tokens and $8 for every one million output tokens, while the pricing is $1.10 for every one million input tokens and $4.40 for one million output tokens for o4-mini. Here we report in Table 7 the per-instance average API cost based on five runs. The remaining components of the system, including LKH initialization and Concorde subproblem solving, are open-source CPU-based tools and therefore incur negligible cost relative to VLM inference. The cost of API calls for individual instances varies due to the number of valid iterations needed to reach the convergence. From djs1000 to pla85900, the average cost ranges between $0.12 and $39.40.

Table 7: Average API cost per instance.

| dsj1000 | pr1002 | u1060 | vm1084 | pcb1173 | d1291 | rl1304 | rl1323 | nrw1379 | fl1400 | u1432 |
|---|---|---|---|---|---|---|---|---|---|---|
| $0.12 | $0.26 | $0.15 | $0.15 | $0.51 | $0.83 | $0.13 | $0.35 | $0.55 | $0.21 | $0.22 |

| fl1577 | d1655 | vm1748 | u1817 | rl1889 | d2103 | u2152 | u2319 | pr2392 | pcb3038 | fl3795 |
|---|---|---|---|---|---|---|---|---|---|---|
| $0.29 | $0.55 | $0.35 | $0.77 | $0.24 | $0.36 | $1.21 | $0.54 | $0.70 | $1.40 | $0.89 |

| fnl4461 | rl5915 | rl5934 | pla7397 | rl11849 | usa13509 | brd14051 | d15112 | d18512 | pla33810 | pla85900 |
|---|---|---|---|---|---|---|---|---|---|---|
| $0.56 | $1.08 | $1.20 | $1.36 | $0.90 | $1.57 | $3.44 | $5.72 | $5.62 | $11.47 | $39.40 |

## J    COMPLETE PLOTS FOR ABLATION STUDIES OF DIFFERENT SELECTION POLICIES

In Figure 7, we illustrate the optimality gap reduction curves among *ViTSP* and two random selectors across 33 TSPLIB instances to demonstrate the effectiveness of VLM in selecting meaningful subproblems.

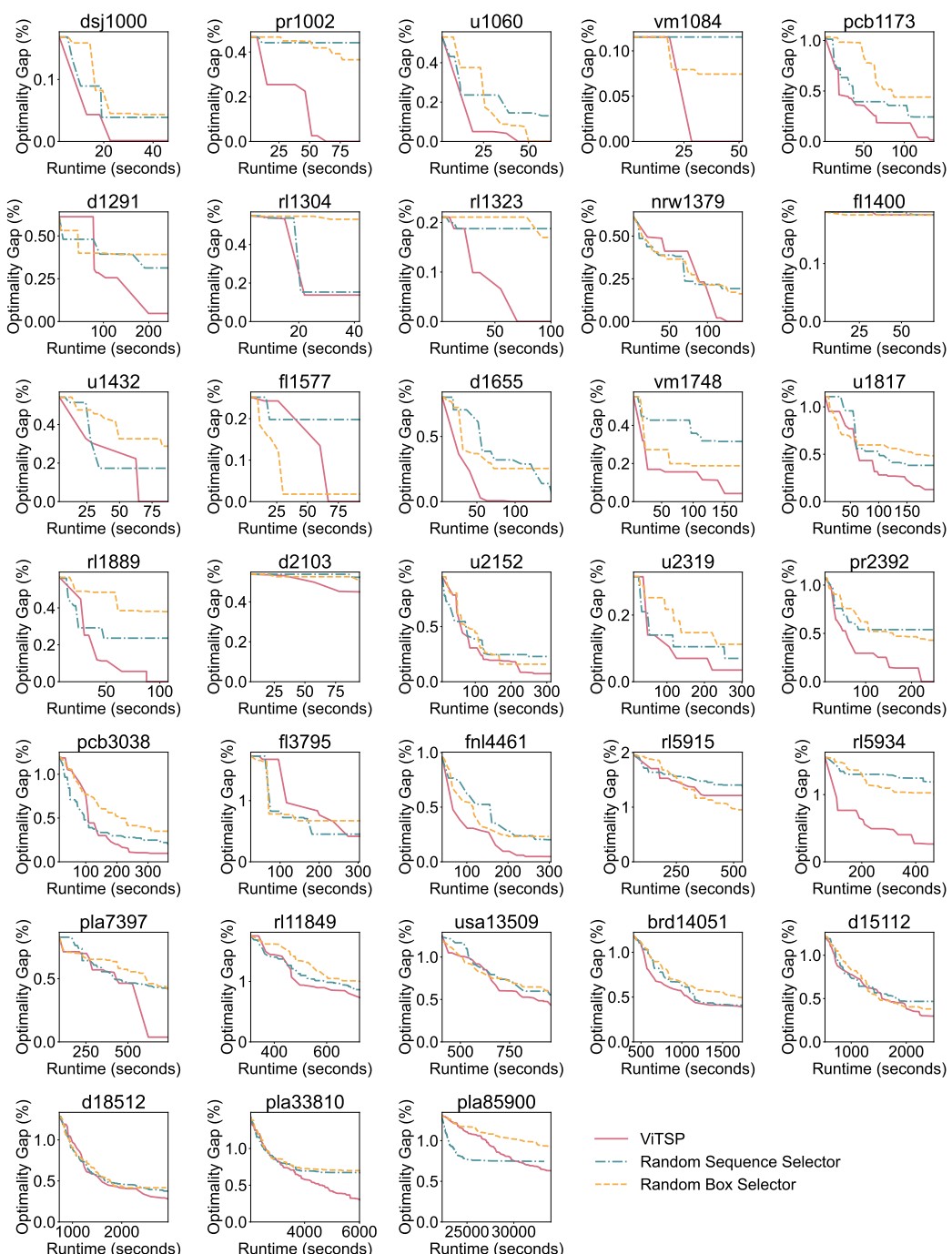

Figure 7: Optimality gap reduction over time among three selection policies.

## K    LIMITATIONS AND BROADER IMPACTS

We discuss three limitations and additional directions of future work in this section. First, the box-based subproblem decomposition guided by VLMs in this study represents one type of metaheuristic operation for combinatorial optimization. As ablation studies show, selecting a sequence of nodes may also be a helpful metaheuristic operation. Thus, exploring additional operations designed by VLMs could further unlock the potential of hybridizing machine learning and operations research methods. Second, although parallel computing is employed, this study does not explicitly optimize the coordination between the selector and solver modules. In particular, there is an unexplored trade-off between solving a single large subproblem with longer runtime versus solving multiple smaller subproblems within the same time. We leave this investigation for future work. Third, exploring batch-parallel execution to enable the optimization of multiple instances simultaneously could further improve overall throughput in practice.

Broadly, the TSP is more than an academic challenge—it is a foundational problem with broad applications across industries, such as transportation, logistics, chip design, and DNA sequencing. This work creates new opportunities for the machine learning community to develop high-quality solutions for large-scale TSP and related problems using general-purpose large models. As VLMs become increasingly accessible and capable of advanced reasoning, they offer scalable solutions deployable on both cloud and edge devices, paving the way for practical and impactful applications.

## L    THE USE OF LARGE LANGUAGE MODELS (LLMS)

The authors confirm that LLMs were used only for grammar checking and text polishing. They were not involved in research ideation. Their role in writing was limited, such that they are not considered contributors.

