# OpenReview forum: "ViTSP: A Vision Language Models Guided Framework for Solving Large-Scale Traveling Salesman Problems"
_ICLR.cc/2026/Conference — ICLR 2026 Poster_

### Official Review · Reviewer_ThrP · 2025-10-15

**Soundness:** 2
**Presentation:** 3
**Contribution:** 2
**Rating:** 4
**Confidence:** 4

**Summary:**

The authors propose a divide-and-conquer approach for large-scale TSP: a VLM selects regions to optimize on a visualization of the current tour, and those regions are then handed to a traditional solver.

**Strengths:**

Introducing LMs into combinatorial optimization is an interesting idea, and the writing is clear.

**Weaknesses:**

1. Limited novelty
In neural TSP solvers, divide-and-conquer is already a common paradigm. The use of a VLM here is not particularly fancy; it mainly selects axis-aligned rectangles and relies on sparse optimization rewards, amounting to a straightforward integration of a VLM into a standard divide-and-conquer framework.

2. Experimental results and setup lack persuasiveness
The method is built on traditional solvers but does not compare against improved neural solvers such as NeuroLKH. The current results only show that the combination of divide-and-conquer framework and efficient solvers (LKH, Concorde) outperforms constructive neural solvers, weak local reform (neuro)solvers, and traditional solvers; they do not demonstrate the added value of introducing a complex VLM.
The benchmarks are mainly several TSPLIB instances, without tests on the commonly used ([0,1]×[0,1]) randomly generated Euclidean datasets. According to the NeuroLKH report, LKH and NeuroLKH can achieve sub–one-per-thousand gaps on such datasets. It would be useful to see how the proposed method performs under the same setting.
The authors are encouraged to discuss whether the method can be applied to more complex, more constrained tasks and to provide attempts. Although focusing on large-scale TSP already represents substantial effort, more comprehensive discussion and trials would be valuable.

Overall, my initial suggestion is borderline reject.

**Questions:**

Please see weaknesses.

---

> ### Author Response · Authors · 2025-12-03
> **Response to Q1**
>
> Thank you for the insightful feedback. Below, we provide detailed responses and additional experiment results to your comments. We hope these clarifications effectively address your concerns.  If there are any remaining issues, we would greatly value further suggestions and are committed to refining the paper accordingly.
>
>
> 1.	Limited novelty in neural TSP solvers, divide-and-conquer is already a common paradigm. The use of a VLM here is not particularly fancy; it mainly selects axis-aligned rectangles and relies on sparse optimization rewards, amounting to a straightforward integration of a VLM into a standard divide-and-conquer framework.
>
> **Response:**
>
> Thank you for your comment. We acknowledge that divide-and-conquer is a long-standing paradigm in classical optimization and is also adopted by existing learning-based local improvement methods such as H-TSP[1], UDC[2], and SO[3]. However, the novelty of our work does not lie in reintroducing divide-and-conquer itself, but rather in demonstrating that pretrained, instruction-following VLMs can serve as effective decomposition heuristics for very-large-scale TSP, a setting where existing neural approaches with dedicated training fail to operate.
>
> As we summarized in the general response, the core contributions that distinguish ViTSP from prior works are:
>
> (1) Multimodal decomposition heuristics from VLMs without dedicated training;
>
> (2) A new hybrid generative AI-OR pipeline enabling problem sizes where neural solvers struggle;
>
> (3) Fair experimental comparison among benchmarks to demonstrate ViTSP’s performance. Notably, we designed an asynchronous orchestration scheme between VLM and subproblem solver to significantly reduce the overall runtime per instance.
>
> This shows that integrating VLM-guided local refinement with exact subproblem solving is not merely “straightforward,” but introduces a previously unexplored path for scaling TSP with general-purpose multimodal models.
>
>
> [1] H-TSP: Hierarchically Solving the Large-Scale Traveling Salesman Problem. AAAI 2023
>
> [2] UDC: A Unified Neural Divide-and-Conquer Framework for Large-Scale Combinatorial Optimization Problems. NeurIPS 2024
>
> [3] Select and Optimize: Learning to solve large-scale TSP instances. AISTATS 2023

---

> ### Author Response · Authors · 2025-12-03
> **Response to Q2**
>
> 2.	Experimental results and setup lack persuasiveness The method is built on traditional solvers but does not compare against improved neural solvers such as NeuroLKH. The current results only show that the combination of divide-and-conquer framework and efficient solvers (LKH, Concorde) outperforms constructive neural solvers, weak local reform (neuro)solvers, and traditional solvers; they do not demonstrate the added value of introducing a complex VLM. The benchmarks are mainly several TSPLIB instances, without tests on the commonly used ([0,1]×[0,1]) randomly generated Euclidean datasets. According to the NeuroLKH report, LKH and NeuroLKH can achieve sub–one-per-thousand gaps on such datasets. It would be useful to see how the proposed method performs under the same setting. The authors are encouraged to discuss whether the method can be applied to more complex, more constrained tasks and to provide attempts. Although focusing on large-scale TSP already represents substantial effort, more comprehensive discussion and trials would be valuable.
>
> **Response:**
>
> Thank you for your insightful suggestion. TSPLIB actually represents more challenging and richer out-of-distribution instances compared to the synthetic random uniform dataset. Most learning-based methods deliver good results, but always fail in TSPLIB. Our experiments show that learning-based methods suffer from solving TSPLIB effectively.
>
> We demonstrated the added value of incorporating VLMs by comparing ViTSP against LKH-3 and Concorde under aligned runtime conditions—a comparison that most prior works did not conduct (see Contribution #3 in the general response). We also provide an anytime gap-reduction analysis between ViTSP and standalone LKH-3 with extended runtime in Figures 2 & 5. These results show that **VLMs contribute effective decomposition heuristics to speed up the local improvement compared to running standalone LKH-3 and Concorde when additional runtime is available.**
>
> We have also added additional experiments to incorporate NeuroLKH as a baseline as we respond to Reviewer AY9B in Q3. However, even though it shows that NeuroLKH is very effective, no methods dominate the solving across all 33 TSPLIB instances.
>
> Regarding testing on synthetic TSP instances, we further run ViTSP on 16 randomly uniformly distributed TSP instances with 10,000 nodes. We also align the runtime by setting the time budget as 600s (10min) as indicated in INViT and compare the average obj and gap with benchmarks. The results further confirm the effectiveness of our proposed ViTSP on very-large-scale TSP, matching with results for TSPLIB instances with n>10,000. Notably, ViTSP outperforms standalone LKH-3 and other learning-based methods.
>
> | Method               | Obj.   | Gap     | Time (s) |
> |----------------------|--------|---------|----------|
> | Near-Optimality     | 71.78  | —       | —        |
> | LKH-3 (more RUNs)    | 72.54  | 1.06%   | 645.0    |
> | FI                   | 80.59  | 12.27%  | 55.3     |
> | AM(G)                | 141.68 | 97.38%  | 17.5     |
> | DIFUSCO (S + 2-opt)  | 73.89  | 2.94%   | 610.0    |
> | INViT                | 76.09  | 6.01%   | 30.9     |
> | SIT (PRC,1000)       | 73.08  | 1.81%   | 1020.0   |
> | DeepACO              | 79.76  | 11.12%  | 605.1    |
> | SO                   | N/A    | —       | —        |
> | UDC                  | OOM    | —       | —        |
> | ViTSP                | **72.28**  | **0.70%** | 615.7    |
>
> For more complex tasks, the proposed VLM-based framework is not limited to TSP. Its visual-prompting mechanism can be readily extended to richer routing tasks by encoding task-specific information into the visual representation. For example, in CVRP, different vehicle routes can be visualized using distinct colors, and customer demands can be represented through node sizes or labels. However, we would like to highlight that the main focus of this paper is TSP, since solving very-large-scale out-of-distribution TSP is itself a substantial undertaking. In contrast, existing literature shows that instance sizes beyond a few thousand and out-of-distribution nodes quickly exceed the practical limits of most learning-based or visual methods. Demonstrating strong performance on such large, real-world TSP instances, therefore, represents a significant advancement relative to prior work.

---

### Official Review · Reviewer_v67N · 2025-10-26

**Soundness:** 3
**Presentation:** 2
**Contribution:** 3
**Rating:** 6
**Confidence:** 2

**Summary:**

The paper tackles the classical TSP with a hybrid approach combining an exact solver (Concorde) with a Vision Language Model. The workflow proceeds iteratively between the solver and the VLM, with the VLM identifying sub problems based on a visualization of the problem + current solution, and the solver improving the current solution. Results show improvements on the TSPLIB benchmark.

**Strengths:**

I think that the paper's idea is original and novel. I am not aware of such an approach combining VLMs and exact solvers for this problem before. I think that the fact that a simple visualization of the current TSP solution being fed to a VLM gives us some meaningful results is pretty interesting and creative, if true!

I think the results presented here seem very promising and significant, as the scale of TSPLIB problems solved by this method is claimed to be unprescedented.

However, I would like to note to the AC that I am not very confident about my review and understanding of the current landscape of this field. I will be reading other reviews carefully during the rebuttal period to identify potential gaps in my assessment.

**Weaknesses:**

I think that presentation of the results can be improved. The paper requires a lot of flipping between the main and appendix. The authors may try to make the main section more self contained when revising the paper.

I also think that, practically, the real cost of a VLM-based approach will likely far exceed Concorde/other solvers at the moment. The authors noted simply "In ViTSP, the VLMs were accessed on demand online. Their usage did not rely on local GPU resources but was confined by the I/O rate." -- but perhaps a more nuanced discussion/justification/cost estimation would be helpful. There is a cost to each API call to an OpenAI-hosted model. Beyond that, there is a cost to running these models themselves.

**Questions:**

1. What is the advantage of ViTSP over Concorde? You claimed that exact solvers like Concorde are become intractable for very large problem sizes -- can you show / have you already shown any problem where Concorde is intractable and ViTSP can do the job?

2. Is it possible to further ablate the VLM component? For instance, how important is the visual prompt vs. the textual prompt?

---

> ### Author Response · Authors · 2025-12-03
> **Response to Q1, Q2 and Q3**
>
> Thank you for the thoughtful feedback. We hope our responses below resolve your concerns. We welcome further feedback and are fully committed to improving the paper.
>
> 1.	What is the advantage of ViTSP over Concorde? You claimed that exact solvers like Concorde are become intractable for very large problem sizes -- can you show / have you already shown any problem where Concorde is intractable and ViTSP can do the job?
>
> **Response:**
>
> Yes, in our original experiments, we have already evaluated ViTSP and Concorde on very-large-scale TSPLIB instances, specifically pla33810 (33,810 nodes) and pla85900 (85,900 nodes).
>
> The key advantage of ViTSP lies in its ability to provide VLM-guided decomposition heuristics without dedicated training, enabling a local-improvement strategy that remains scalable even when exact solvers become computationally infeasible.
>
> Under our hardware configuration (AMD EPYC 7443 24-Core CPU), Concorde reported failure when working on both instances, whereas ViTSP successfully produced high-quality solutions, including improvements over LKH-3 under the same runtime budget. These results demonstrate that ViTSP can remain effective in problem regimes where Concorde becomes intractable.
>
> ****
>
> 2.	Is it possible to further ablate the VLM component? For instance, how important is the visual prompt vs. the textual prompt?
>
> **Response:**
>
> Thank you for this valuable question. In ViTSP, the visual and textual prompts play complementary and essential roles in eliciting effective subproblem selection from the VLM. The visual prompt captures the geometric structure of the TSP instance—spatial distribution, clustering patterns, and locality—which is crucial for identifying meaningful regions. The textual prompt specifies the task objective (e.g., selecting promising subregions) and provides the semantic grounding that guides the model’s reasoning about the visual input.
>
> This design is motivated by limitations observed in prior work. Approaches that rely solely on textual input—such as LLM-as-optimizer and EoH methods that take only coordinate lists—struggle to produce high-quality solutions even for relatively small problems (e.g., TSP with n = 50). Conversely, prior multimodal approaches that provide only visual input and attempt to read node indices from images are unable to scale effectively, as visual clutter and index ambiguity grow rapidly with problem size.
>
> We have added a discussion to the revised manuscript in section 4.4 lines 537-543 to clarify the contribution of each component.
>
> ***
>
> 3.	I think that presentation of the results can be improved. The paper requires a lot of flipping between the main and appendix. The authors may try to make the main section more self contained when revising the paper.
>
> **Response:**
>
> We agree that consolidating key results in the main text can improve readability. Due to page limits, many detailed figures were moved to the appendix, with Table 1 serving as the central summary of our core findings in the main section. In the revision, we have added additional clarifications and experiments in the main section based on the comments from reviewers to make the main section more self-contained.

---

> ### Author Response · Authors · 2025-12-03
> **Response to Q4**
>
> 4.	I also think that, practically, the real cost of a VLM-based approach will likely far exceed Concorde/other solvers at the moment. The authors noted simply "In ViTSP, the VLMs were accessed on demand online. Their usage did not rely on local GPU resources but was confined by the I/O rate." -- but perhaps a more nuanced discussion/justification/cost estimation would be helpful. There is a cost to each API call to an OpenAI-hosted model. Beyond that, there is a cost to running these models themselves.
>
> **Response:**
>
> Thank you for this insightful comment. We agree that a more nuanced discussion of the practical cost of VLM-based approaches is important. In ViTSP, the primary additional cost indeed comes from online VLM API calls, and we report the per-instance API cost in the manuscript to make this overhead transparent. The remaining components of the system—LKH initialization and Concorde subproblem solving—are open-source CPU-based tools and therefore incur negligible cost relative to VLM inference. From djs1000 to pla85900, the average cost based on five runs ranges from \$0.12 \text{to} \$39.40. We have added a subsection "Cost of VLM API calls per instance" in Section 4.4 to reflect this cost analysis.
>
> | Instance | dsj1000 | pr1002 | u1060 | vm1084 | pcb1173 | d1291 | rl1304 | rl1323 | nrw1379 | fl1400 | u1432 |
> |----------|---------|--------|--------|---------|---------|--------|--------|--------|---------|--------|--------|
> | Cost     | $0.12   | $0.26  | $0.15 | $0.15   | $0.51   | $0.83  | $0.13  | $0.35  | $0.55   | $0.21  | $0.22  |
>
> | Instance | fl1577 | d1655 | vm1748 | u1817 | rl1889 | d2103 | u2152 | u2319 | pr2392 | pcb3038 | fl3795 |
> |----------|--------|--------|--------|--------|--------|--------|--------|--------|--------|---------|--------|
> | Cost     | $0.29  | $0.55  | $0.35  | $0.77  | $0.24  | $0.36  | $1.21  | $0.54  | $0.70  | $1.40   | $0.89  |
>
> | Instance | fnl4461 | rl5915 | rl5934 | pla7397 | rl11849 | usa13509 | brd14051 | d15112 | d18512 | pla33810 | pla85900 |
> |----------|----------|--------|---------|----------|----------|-----------|-----------|--------|--------|-----------|-----------|
> | Cost     | $0.56     | $1.08  | $1.20    | $1.36     | $0.90     | $1.57      | $3.44      | $5.72   | $5.62   | $11.47     | $39.40     |

---

### Official Review · Reviewer_ZmWq · 2025-10-28

**Soundness:** 2
**Presentation:** 3
**Contribution:** 2
**Rating:** 4
**Confidence:** 4

**Summary:**

This paper proposes a vision-guided solution framework ViTSP for Traveling Salesman Problem, leveraging pre-trained VLMs to act as selector of improvement region and exact solvers to settle down the subproblem. The authors also conduct experiments on some real-world instances in TSPLIB to show the performance on large-scale scenario.

**Strengths:**

1 This paper introduce VLM to directly determine the box region to be revised instead of time-consuming training, which is an interesting attempt compared to traditional learning-based paradigm.

2 The proposed method outperforms the baselines on most of large-scale non-uniform real-world instances (N>1000).

3 Figure 1 presents clear and nice visualization to understand the pipline.

**Weaknesses:**

1 The experiment is not sufficient for merely certain large-scale instances in TSPLIB. I suggest that the authors need to demonstrate its effectiveness on other random uniform and non-uniform datasets, datasets in INViT.

2 ViTSP is essentially an improved approach. So how does its iterative improvement efficiency compare to other neural improvement methods?

3 Dealing with TSP in a visual way is not a new
thing. In the past, there were a series of methods that used two-dimensional scatter plots as input. I suggest the author review relevant visual methods and elaborate on the advantages of VLM compared to them.

4 The way to leverage VLM in this paper is invoking API, which is a simple way to use a large model. But is it possible to design a reasonable way to fine tune the model based on TSP tasks, making the combination of VLM and tasks more collaborative?

5 The problem discussed in this paper is merely TSP, which is narrow. Can the VLM-based method adapt to solving other routing tasks?

6 What is the initialization method used in this paper? From the title of Figure 4 in Appendix, it appears to be LKH3. Is the parameter values (especially the number of runs) of LKH-3 in initialization default? What's more, the quality of iteration and final solution highly depend on initialization. How about the performance of VLM-based improvement integrated with other solution initialization methods, e.g. random initialization or insertion initialization? I suggest the author provide more details about de initialization and comparative experiments with other initialization methods.

7 Does the solution time include the initial solution generation time? I suggest the author provide more details about the solution time, including initial solution generation, VLM inference, and exact solving of subproblems.

8 Does the input 2D image in ViTSP need scale, quality, and formatting requirements? When the instance scale is huge, scatter plots and circumnavigation maps appear crowded and convergent visually, which leads to a serious lack of fine-grained information. Applying the same scaling to instances of different sizes does not seem to lead to good visual perception of VLM. How was this paper handled?

9 The leverage of VLM avoids the cost of intensive training. However, I would like to know if it would cause an increase in inference time compared to end-to-end methods.

10 It seems that ViTSP is iteratively improved instance by instance. Can it be executed in parallel in batches?

**Questions:**

Please see weaknesses. The idea is very interesting and  I will adjust the paper’s score accordingly in my rebuttal to reflect the strengths identified.

---

> ### Author Response · Authors · 2025-12-03
> **Response to Q1, Q2, and Q3**
>
> We sincerely thank the Reviewer for finding our work interesting and for providing constructive feedback. We address each of your comments in detail below and hope that our responses have adequately resolved your concerns.
>
> 1.	The experiment is not sufficient for merely certain large-scale instances in TSPLIB. I suggest that the authors need to demonstrate its effectiveness on other random uniform and non-uniform datasets, datasets in INViT.
>
> **Response:**
>
> Thank you for your insightful suggestion. We have further run ViTSP on 16 randomly uniformly distributed TSP instances with 10,000 nodes as in INViT. We also aligned the runtime by setting the time budget as 600s (10min) as indicated in INViT and compared the average obj and gap with benchmarks. The results further confirm the effectiveness of our proposed ViTSP on very-large-scale TSP, aligning with the superior performance of ViTSP in TSPLIB instances with n>10,000. Notably, ViTSP outperforms standalone LKH-3 and other learning-based methods. This additional experiment is added as Table 3 in Section 4.2 in the revised manuscript.
>
> | Method               | Obj.   | Gap     | Time (s) |
> |----------------------|--------|---------|----------|
> | Global Optimality     | 71.78  | —       | —        |
> | LKH-3 (more RUNs)    | 72.54  | 1.06%   | 645.0    |
> | FI                   | 80.59  | 12.27%  | 55.3     |
> | AM(G)                | 141.68 | 97.38%  | 17.5     |
> | DIFUSCO (S + 2-opt)  | 73.89  | 2.94%   | 610.0    |
> | INViT                | 76.09  | 6.01%   | 30.9     |
> | SIT (PRC,1000)       | 73.08  | 1.81%   | 1020.0   |
> | DeepACO              | 79.76  | 11.12%  | 605.1    |
> | SO                   | N/A    | —       | —        |
> | UDC                  | OOM    | —       | —        |
> | ViTSP                | 72.28  | 0.70%   | 615.7    |
>
> ***
> 2.  ViTSP is essentially an improved approach. So how does its iterative improvement efficiency compare to other neural improvement methods?
>
> **Response:**
>
> Thank you for your question regarding the efficiency of iterative improvement. Neural improvement baselines (e.g., DeepACO, SO, UDC) and our ViTSP framework adopt heterogeneous settings in terms of initialization, selection strategies, and refinement policies, which makes a strict one-to-one comparison of per-iteration efficiency less meaningful. For instance, SO refines an open TSP instance using a trained neural solver, whereas ViTSP reformulates a subproblem into a standard TSP and then leverages Concorde to solve each subproblem to optimality.
>
> Given these structural differences, we follow prior works and compare efficiency at the overall performance level under the same (or even longer for DeepACO) runtime budgets. Under comparable wall-clock time, ViTSP consistently achieves smaller optimality gaps than existing neural improvement methods, demonstrating superior overall improvement efficiency despite differences in their internal iteration mechanics.
>
> ***
> 3.	Dealing with TSP in a visual way is not a new thing. In the past, there were a series of methods that used two-dimensional scatter plots as input. I suggest the author review relevant visual methods and elaborate on the advantages of VLM compared to them.
>
> **Response:**
>
> Thank you for this helpful suggestion. Our literature review has primarily focused on learning-based TSP solvers and classical OR techniques since the surge of generative AI. One of the most relevant works is [1], which attempts to read node indices directly from the plot. These designs do not scale well: as the instance size grows, node indices become visually cluttered, and the model struggles to identify meaningful geometric structure, leading to degraded performance on large TSPLIB instances.
> The key advantage of VLM compared to prior works is that we leveraged the broad multimodal pretraining of modern VLMs, which enables VLMs to understand TSP-specific prompts and reason about geometric layouts without any supervised or reinforcement training on TSP. This allows our method to intelligently identify promising subregions even for very large instances, which is an ability that prior scatter-plot–based visual methods do not demonstrate.
>
> If there are specific visual TSP works that the reviewer believes should be discussed, we would greatly appreciate any pointers so that we can include them in the updated related-work section.
>
> [1] Visual reasoning and multi-agent approach in multimodal large language models (mllms): Solving tsp and mtsp combinatorial challenges. arXiv preprint arXiv:2407.00092.

---

> ### Author Response · Authors · 2025-12-03
> **Response to Q4 and Q5**
>
> 4.	The way to leverage VLM in this paper is invoking API, which is a simple way to use a large model. But is it possible to design a reasonable way to fine tune the model based on TSP tasks, making the combination of VLM and tasks more collaborative?
>
> **Response:**
>
> Thank you for raising this thoughtful point regarding model fine-tuning. Our goal in this work is to pioneer the use of pretrained VLMs as zero-shot decomposition heuristics for large-scale TSP, without any task-specific training. This is a core advantage of ViTSP for general users: it allows our approach to remain general-purpose, easily reproducible, and avoid the extensive GPU resources required by existing learning-based combinatorial optimization models. Our study has demonstrated that even without fine-tuning, pretrained VLMs can meaningfully guide decomposition for very-large-scale TSP.
>
> Fine-tuning VLMs on TSP tasks is indeed a promising future direction for building deeper collaboration between the model and the solver. Recent work, such as [2], suggests that instruction-aligned models can improve CO performance when heavily fine-tuned. However, such methods typically require substantial GPU budgets, specialized datasets, and nontrivial engineering, which places them closer to fully trained neural solvers rather than general heuristics.
>
> To acknowledge the possibility of fine-tuning VLMs for routing tasks, we have added it as a future work direction in the Conclusions section:
>
> > Lightweight fine-tuning of offline VLMs, as well as further exploration of in-context reinforcement learning, represent promising future directions to further enhance performance of VLMs for TSP and broad routing problems.
>
> ***
> 5.	The problem discussed in this paper is merely TSP, which is narrow. Can the VLM-based method adapt to solving other routing tasks?
>
> **Response:**
>
> Thank you for this question about the expansion of our work. We have discussed this point in our original paper in the Conclusions section.
>
> >Because TSP serves as the base case for a wide family of routing problems, the promising results from ViTSP suggest opportunities to expand our framework to other routing problems.
>
> The proposed VLM-based framework is not limited to TSP. Its visual-prompting mechanism can be readily extended to richer routing tasks by encoding task-specific information into the visual representation. For example, in CVRP, different vehicle routes can be visualized using distinct colors, and customer demands can be represented through node sizes or labels. While this paper focuses on TSP, we would like to highlight that solving very-large-scale out-of-distribution TSP is itself a substantial undertaking, as existing literature shows that instance sizes beyond a few thousand and out-of-distribution nodes quickly exceed the practical limits of most learning-based or visual methods. Demonstrating strong performance on such large, real-world TSP instances, therefore represents a significant advancement relative to prior work.
>
> [2] Large Language Models as End-to-end Combinatorial Optimization Solvers, NeurIPS 2025

---

> ### Author Response · Authors · 2025-12-03
> **Response to Q6 and Q7**
>
> 6.	What is the initialization method used in this paper? From the title of Figure 4 in Appendix, it appears to be LKH3. Is the parameter values (especially the number of runs) of LKH-3 in initialization default? What's more, the quality of iteration and final solution highly depend on initialization. How about the performance of VLM-based improvement integrated with other solution initialization methods, e.g. random initialization or insertion initialization? I suggest the author provide more details about de initialization and comparative experiments with other initialization methods.
>
> **Response:**
>
> Yes, we use LKH-3 with default parameters as our initialization method. The purpose of selecting default LKH-3 is to avoid the parameter calibration per instance that relies on domain-specific knowledge.
>
> We have updated Section 4.3 to add this ablation study. Here we report the performance of ViTSP, initialized by Farthest insertion (FI) and LKH-3 (default), respectively. The ViTSP initialized by LKH consistently outperforms the one initialized by FI. These results confirm that while initialization affects performance, as expected, the VLM-guided decomposition consistently improves over its corresponding initialization baseline.
>
> | ViTSP             | dsj1000 | pr1002 | u1060 | vm1084 | pcb1173 | d1291 | rl1304 | rl1323 | nrw1379 | fl1400 | u1432 |
> |--------------------|---------|--------|-------|--------|---------|-------|--------|--------|---------|--------|--------|
> | runtime (s)        | 69.6    | 65.0   | 57.1  | 38.5   | 124.3   | 168.6 | 27.2   | 95.6   | 104.6   | 53.9   | 83.2   |
> | LKH initialization | 0.02%   | 0.01%  | 0.00% | 0.00%  | 0.16%   | 0.15% | 0.14%  | 0.04%  | 0.00%   | 0.18%  | 0.03%  |
> | FI initialization  | 0.50%   | 0.03%  | 0.64% | 0.67%  | 0.21%   | 2.42% | 8.01%  | 0.39%  | 0.01%   | 1.30%  | 1.43%  |
>
> | ViTSP             | fl1577 | d1655 | vm1748 | u1817 | rl1889 | d2103 | u2152 | u2319 | pr2392 | pcb3038 | fl3795 |
> |--------------------|--------|-------|--------|-------|--------|-------|--------|--------|---------|----------|---------|
> | runtime (s)        | 58.9   | 111.2 | 162.8  | 180.0 | 89.0   | 84.0  | 238.3  | 244.9  | 187.6   | 336.3    | 252.4   |
> | LKH initialization | 0.00%  | 0.00% | 0.05%  | 0.24% | 0.03%  | 0.39% | 0.08%  | 0.06%  | 0.09%   | 0.09%    | 0.57%   |
> | FI initialization | 17.61% | 6.56%  | 0.04% | 3.16%  | 0.50% | 23.55% | 0.98% | 0.07%  | 0.41%  | 0.34%   | 3.02%    |
>
> | ViTSP            | fnl4461 | rl5915 | rl5934 | pla7397 | rl11849 | usa13509 | brd14051 | d15112 | d18512 | pla33810 | pla85900 |
> |--------------------|---------|--------|--------|---------|----------|-----------|-----------|--------|--------|-----------|-----------|
> | runtime (s)        | 277.3   | 386.5  | 485.5  | 575.5   | 785.1    | 995.2     | 1725.3    | 2172.0 | 2319.0 | 5758.1    | 29863.6   |
> | LKH initialization | 0.16%   | 1.13%  | 0.41%  | 0.28%   | 1.03%    | 0.47%     | 0.31%     | 0.22%  | 0.36%  | 0.52%     | 0.83%     |
> | FI initialization  |  2.49%  |8.42%   | 6.40%  | 5.72%  | 9.36%   | 8.89%    | 0.52%     | 0.24%     | 0.54%  | 10.65% | 2.81%     |
>
>
> ***
> 7.	Does the solution time include the initial solution generation time? I suggest the author provide more details about the solution time, including initial solution generation, VLM inference, and exact solving of subproblems.
>
> **Response:**
>
> Yes, our study reports fully aligned and fair runtime measurements. We have revised our manuscript to explain the solution time in lines 322-323.
>
> >The recorded wall-clock solution time of ViTSP explicitly includes: (1) the LKH-3 initialization time, (2) all VLM API inference latency and waiting time, and (3) the Concorde subproblem solving time.
>
> The initial solution generation time is identical to the default LKH-3 (Default) runtime, as we use LKH-3 with its standard parameter configuration for initialization. After initialization, VLM inference and exact subproblem solving proceed asynchronously, and the remaining runtime budget is allocated to both components. Their anytime behavior following initialization is shown in Figures 2 & 5, which illustrates the iterative improvements contributed by the VLM-guided decomposition.

---

> ### Author Response · Authors · 2025-12-03
> **Response to Q8, Q9, and Q10**
>
> 8.	Does the input 2D image in ViTSP need scale, quality, and formatting requirements? When the instance scale is huge, scatter plots and circumnavigation maps appear crowded and convergent visually, which leads to a serious lack of fine-grained information. Applying the same scaling to instances of different sizes does not seem to lead to good visual perception of VLM. How was this paper handled?
>
> **Response**:
>
> Thank you for this question. We have modified Appendix C to better explain this detail.
>
> > In our implementation, all TSP instances are rendered using a consistent figure size (e.g., figsize = (20, 20)).
>
> While extremely large instances naturally generate crowded scatter plots, ViTSP handles this challenge through the zoom-in selection mechanism described in Section 3.3. Specifically, when the global plot becomes visually congested, for example, in instances with tens of thousands of nodes, the VLM is first asked to identify coarse regions of interest. ViTSP then generates zoomed-in subimages of these regions with figsize=(20,20), where nodes are visually separated and edge structures become clearer. The VLM performs a second round of selection on these high-resolution local views, ensuring that its perception remains reliable even when the full-instance image lacks fine-grained detail.
>
> ***
>
> 9.	The leverage of VLM avoids the cost of intensive training. However, I would like to know if it would cause an increase in inference time compared to end-to-end methods.
>
> **Response:**
>
> Thank you for this question. VLM inference is slower per call due to large model inference latency and online API I/O latency.  In practice, an OpenAI VLM call ranges from 3–5 s for “fast-thinking” inference and can exceed 10 s for “reasoning” modes. However, there is an inherently schematic difference between ViTSP and end-to-end methods: end-to-end methods perform a one-time forward pass through a comparatively small neural network. While its overall one-time forward inference time is short, its solution generation process is completed. On the other hand, ViTSP is a local improvement framework in which VLM offers decomposition heuristics to consistently improve the solution, instead of constructing end-to-end full solutions.
>
> ***
> 10.	It seems that ViTSP is iteratively improved instance by instance. Can it be executed in parallel in batches?
>
> **Response:**
>
> Thank you for this question. While ViTSP is presented as an iterative improvement framework operating on each instance independently, as is also the case for most existing improvement-based TSP solvers, it can indeed be executed in parallel when computing resources allow. The asynchronous orchestration between VLM selection and Concorde subproblem solving is designed based on single instances, but nothing in the framework prevents running multiple instances simultaneously. For example, ViTSP can process a batch of instances by parallelizing the LKH initialization, distributing VLM inference across multiple available API channels, and executing Concorde subproblems concurrently on multi-core CPUs. In addition, Concorde can also solve subproblems from different instances in sequence while waiting for VLM-guided subproblem selections from a particular instance.
>
> We have added a brief discussion of this capability in Appendix J of the revised manuscript as follows:
> > Exploring batch-parallel execution to enable the optimization of multiple instances simultaneously could further improve
> overall throughput in practice.

---

### Official Review · Reviewer_RptF · 2025-10-31

**Soundness:** 3
**Presentation:** 3
**Contribution:** 3
**Rating:** 6
**Confidence:** 5

**Summary:**

This paper proposes ViTSP, a hybrid optimization framework that leverages pre-trained Vision-Language Models (VLMs) to identify promising spatial subregions in large-scale Traveling Salesman Problem (TSP) instances. These subproblems are then optimized by Concorde, forming an asynchronous pipeline with LKH-3 used for initialization. The approach avoids model training, claims strong scalability, and achieves competitive optimality gaps on 33 TSPLIB instances up to 85k nodes, outperforming LKH-3 under equal runtime budgets in many cases and achieving up to 0.19% average optimality gap.

**Strengths:**

The framework is well-designed and modular (visual selection, subproblem reconstruction, exact solver, asynchronous orchestration).

Extensive experiments and detailed comparisons on large-scale real benchmarks (TSPLIB, N ≥ 1000) demonstrate practical effectiveness.

No need for training/fine-tuning; low deployment barrier (uses existing VLM APIs + classical solvers), aligning well with real-world engineering needs.

**Weaknesses:**

Insufficient quantitative analysis on VLM output stability, repeatability, and prompt sensitivity.

Fairness of runtime alignment and whether real online VLM latency is fully included needs clearer justification.

Lacks theoretical analysis (convergence/performance guarantees) and interpretability evidence (why VLM can select “beneficial” regions).

**Questions:**

Please provide the full prompt text, the average number of boxes returned per call, and statistics on invalid VLM outputs

For result stability, report mean ± std (or confidence intervals) over 5–10 independent runs per instance to ensure results are not accidental.

Clarify runtime measurement: does the VLM API waiting/latency count toward total runtime?

Regarding baselines: how are LKH-3 (multiple runs) parameters set? How are time limits aligned for Concorde, DeepACO, etc.? Please provide a per-instance configuration table (in appendix or supplementary).


If considering offline/self-hosted VLMs (or fine-tuned models), what is the scalability and cost trade-off? Have the authors tried lightweight fine-tuning or RL signal-guided selection enhancement?

---

> ### Author Response · Authors · 2025-12-03
> **Response to Q1**
>
> Thank you for your review. We provide our responses below to address your questions.
>
> 1.	Please provide the full prompt text, the average number of boxes returned per call, and statistics on invalid VLM outputs
>
> **Response:**
>
> Thank you for the question.
>
> **The full prompt text** was reported in the original manuscript in Appendix C, together with example visual prompts.
>
> **The number of boxes returned per call** is controlled by a parameter Q, as originally described in line 327.
> >We set the number of subproblems generated per prompt Q = 2.
>
> This is because VLMs can be prompted to output user-tailored tasks. We set Q=2 in this study to improve efficiency per call.
>
> **The average valid subproblem selection rate per instance** over 5 runs is reported below and has been added as Table 5 in Section 4.4. The valid rate is the subproblems truly contributed to gap reductions divided by the total subproblems returned from VLMs during the iteration. The average valid rate ranges between 10% to 50%. Notably, these valid rates are achieved without any task-specific training, as the VLMs used are entirely general-purpose.
>
> | Instance | dsj1000 | pr1002 | u1060 | vm1084 | pcb1173 | d1291 | rl1304 | rl1323 | nrw1379 | fl1400 | u1432 |
> |----------|---------|--------|--------|--------|---------|--------|--------|--------|---------|--------|--------|
> | Valid Rate | 41.72% | 53.81% | 48.14% | 35.93% | 35.60% | 10.42% | 41.67% | 30.15% | 39.40% | 33.00% | 34.92% |
>
> | Instance | fl1577 | d1655 | vm1748 | u1817 | rl1889 | d2103 | u2152 | u2319 | pr2392 | pcb3038 | fl3795 |
> |----------|--------|--------|--------|--------|--------|--------|--------|--------|--------|---------|--------|
> | Valid Rate | 48.91% | 25.44% | 34.26% | 24.75% | 31.14% | 50.00% | 22.64% | 12.83% | 37.47% | 30.66%  | 20.50% |
>
> | Instance | fnl4461 | rl5915 | rl5934 | pla7397 | rl11849 | usa13509 | brd14051 | d15112 | d18512 | pla33810 | pla85900 |
> |----------|----------|--------|---------|----------|----------|-----------|-----------|--------|--------|-----------|-----------|
> | Valid Rate | 43.99%  | 23.06% | 37.62%  | 29.05%   | 17.02%   | 25.47%     | 36.84%    | 39.22% | 35.89% | 38.90%    | 40.37%    |
>
> Overall, our results demonstrate that VLMs can generate practically meaningful decomposition heuristics. Parameters such as box size, number of boxes per call, and subregion shape offer additional room for optimization. Future work may also fine-tune VLMs to further improve the valid selection rate.

---

> ### Author Response · Authors · 2025-12-03
> **Response to Q2 and Q3**
>
> 2.	For result stability, report mean ± std (or confidence intervals) over 5–10 independent runs per instance to ensure results are not accidental.
>
> **Response:**
>
> Thank you for the comment. Due to the space limit, we updated the mean gap in Tables 1 & 2 in the revised manuscript to reflect the average results based on 5 independent runs per instance. The detailed mean and std are shown as follows. The results are consistent with what we originally reported, and they are not accidental, confirming that ViTSP is an effective method for solving large-scale TSP.
>
> |                   | dsj1000      | pr1002       | u1060        | vm1084       | pcb1173       | d1291        | rl1304       | rl1323       | nrw1379       | fl1400       | u1432        |
> |-------------------|--------------|--------------|--------------|--------------|---------------|--------------|--------------|--------------|---------------|--------------|--------------|
> | gap               | 0.02% ± 0.02%| 0.01% ± 0.02%| 0.00% ± 0.00%| 0.00% ± 0.00%| 0.16% ± 0.11% | 0.15% ± 0.14%| 0.14% ± 0.00%| 0.04% ± 0.04%| 0.00% ± 0.00% | 0.18% ± 0.00%| 0.03% ± 0.02%|
> | runtime (s)       | 69.6 ± 28.6  | 65.0 ± 26.8  | 57.1 ± 9.8   | 38.5 ± 18.1  | 124.3 ± 29.0  | 168.6 ± 56.9 | 27.2 ± 5.3   | 95.6 ± 24.4  | 104.6 ± 37.6  | 53.9 ± 17.5  | 83.2 ± 16.6  |
>
> |                   | fl1577       | d1655        | vm1748       | u1817        | rl1889       | d2103       | u2152       | u2319       | pr2392        | pcb3038       | fl3795        |
> |-------------------|--------------|--------------|--------------|--------------|--------------|-------------|-------------|-------------|----------------|----------------|---------------|
> | gap               | 0.00% ± 0.00%| 0.00% ± 0.00%| 0.05% ± 0.01%| 0.24% ± 0.08%| 0.03% ± 0.04%| 0.39% ± 0.06%| 0.08% ± 0.01%| 0.06% ± 0.02%| 0.09% ± 0.13% | 0.09% ± 0.01% | 0.57% ± 0.11% |
> | runtime (s)       | 58.9 ± 9.4   | 111.2 ± 18.3 | 162.8 ± 15.9 | 180.0 ± 22.0 | 89.0 ± 19.4  | 84.0 ± 21.7 | 238.3 ± 70.8| 244.9 ± 24.0| 187.6 ± 60.1  | 336.3 ± 26.7  | 252.4 ± 57.4  |
>
> |                   | fnl4461       | rl5915       | rl5934       | pla7397       | rl11849       | usa13509      | brd14051      | d15112       | d18512       | pla33810       | pla85900       |
> |-------------------|---------------|--------------|--------------|---------------|---------------|---------------|---------------|--------------|--------------|------------------|------------------|
> | gap               | 0.16% ± 0.11% | 1.13% ± 0.16%| 0.41% ± 0.16%| 0.28% ± 0.24% | 1.03% ± 0.28% | 0.47% ± 0.06% | 0.31% ± 0.05% | 0.22% ± 0.02%| 0.36% ± 0.10%| 0.52% ± 0.27%   | 0.83% ± 0.18%   |
> | runtime (s)       | 277.3 ± 25.3  | 386.5 ± 78.6 | 485.5 ± 28.9 | 575.5 ± 27.9  | 785.1 ± 85.5  | 995.2 ± 100.0 | 1725.3 ± 42.9 | 2172.0 ± 316.8| 2319.0 ± 785.5| 5758.1 ± 655.2 | 29863.6 ± 2948.1 |
>
> ***
> 3.	Clarify runtime measurement: does the VLM API waiting/latency count toward total runtime? (Fairness of runtime alignment and whether real online VLM latency is fully included needs clearer justification.)
>
> **Response:**
>
> Thank you for emphasizing the importance of runtime measurement. Our runtime reporting is fully aligned and fair across all compared methods. **The online VLM call latency is fully recorded.** This computational overhead also motivates the design of asynchronous orchestration of ViTSP in Section 3.5.
>
> We have added the following details in Section 4.1 - Evaluation metrics:
>
> > The recorded wall-clock runtime of ViTSP explicitly includes : (1) the LKH initialization, (2) all VLM API waiting and latency, and (3) the Concorde solving time for subproblems. Thus, our reported time reflects the actual end-to-end wall time required by ViTSP in real-world settings.
>
> Additionally, as we originally reported in lines 342 - 348, the performances of LKH-3 and Concorde were already reported under an identical wall-clock budget as ViTSP to ensure comparison fairness.
>
> > We align the runtime across baselines to ensure fair comparison. In addition to using the default
> parameter values of LKH-3, we introduce LKH-3 (more RUNS), where the RUNS value is increased
> to match LKH-3’s runtime with that of ViTSP on each instance. Similarly, Concorde, DeepACO,
> UDC, and EoH are run with the same or slightly longer time limit as ViTSP. For FI, runtime is
> deterministic with respect to instance size.

---

> ### Author Response · Authors · 2025-12-03
> **Response to Q4 and Q5**
>
> 4.	Regarding baselines: how are LKH-3 (multiple runs) parameters set? How are time limits aligned for Concorde, DeepACO, etc.? Please provide a per-instance configuration table (in appendix or supplementary).
>
> **Response:**
>
> Thank you for your suggestion. We have updated Appendix F in the revised manuscript to better explain the implementation details.
>
> **LKH-3**. LKH-3’s performance is controlled by parameters like RUNS, TRIALS, etc. By default, RUNS is set to 10, which is used for LKH-3 (default) in our study. TRIALS is always set to be equivalent to the problem size, e.g., TRIALS=1000 for dsj1000. We extend RUNS ad hoc to ensure LKH-3 runtime matches with ViTSP as follows :
>
> |                       | dsj1000 | pr1002 | u1060 | vm1084 | pcb1173 | d1291 | rl1304 | rl1323 | nrw1379 | fl1400 | u1432 | fl1577 |
> |-----------------------|---------|--------|-------|--------|---------|-------|--------|--------|---------|--------|--------|--------|
> | RUNS | 160     | 910    | 750   | 160    | 660     | 1600  | 260    | 360    | 760     | 210    | 410    | 60     |
>
> |                       | d1655 | vm1748 | u1817 | rl1889 | d2103 | u2152 | u2319 | pr2392 | pcb3038 | fl3795 | fnl4461 |
> |-----------------------|-------|--------|-------|--------|-------|--------|--------|---------|----------|---------|----------|
> | RUNS | 560   | 560    | 650   | 310    | 310   | 760    | 460    | 460     | 460      | 310     | 110      |
>
> |                       | rl5915 | rl5934 | pla7397 | rl11849 | usa13509 | brd14051 | d15112 | d18512 | pla33810 | pla85900 |
> |-----------------------|--------|--------|----------|----------|-----------|-----------|---------|---------|-----------|-----------|
> | RUNS | 130    | 130    | 130      | 44       | 30        | 50        | 70      | 50      | 50        | 18        |
>
> **Concorde.** Concorde provides a TIMELIMIT parameter that can be used to constrain its execution time. In our experiments, this timelimit is set to match the per-instance runtime of ViTSP reported in Table 1. Note that Concorde is unable to strictly terminate the process as soon as it reaches the limits.
>
> **UDC.** The UDC solver runs iteratively until it reaches the wall-time limit aligned with ViTSP.
>
> **DeepACO.**  we use the configuration n_ants = 100, n_nodes=500, k_sparse=100, t_aco=100 in the codes. This setting significantly increases the runtime that ViTSP’s runtime but the solution quality remains substantially worse.
>
> ***
>
> 5.	If considering offline/self-hosted VLMs (or fine-tuned models), what is the scalability and cost trade-off? Have the authors tried lightweight fine-tuning or RL signal-guided selection enhancement?
>
> **Response:**
>
> Thank you for your suggestion. We have conducted preliminary experiments with a small offline VLM, specifically Qwen3-VL-8B-Instruct. The performance was substantially worse than the OpenAI-hosted VLMs. We observed two key practical issues:
>
> (1) **Inference latency instability**. The offline model produced inference latency with higher variance. The latency of certain call can be very long, which directly extends the wall-clock time of ViTSP in practice.
>
> (2) **Weaker instruction-following and output formatting reliability**. Despite careful prompting, the offline model frequently produced outputs that deviated from the required bounding-box format. These invalid outputs cause ViTSP to fail to parse the correct subproblems. Based on the definition of invalid output rate in Q1, the invalid output rates are above 95% across instances when using an offline small VLM, which fail to meet the practical use.
>
> Regarding the suggestion of lightweight fine-tuning or RL signal-guided selection, we acknowledge that these are promising directions for future work. However, such approaches fall outside the scope of this paper, which focuses on demonstrating that pre-trained, general-purpose VLMs already possess meaningful zero-shot visual-geometric perception that can be directly leveraged for high-quality TSP decomposition, without any task-specific training.  We have revised the Conclusions section to reflect this.

---

### Official Review · Reviewer_AY9B · 2025-11-01

**Soundness:** 1
**Presentation:** 3
**Contribution:** 1
**Rating:** 2
**Confidence:** 4

**Summary:**

This paper proposes a vision-language-based framework that utilizes VLMs to identify subareas capable of reconstructing trajectories. The use of vision-language models is novel, and the evaluation on large-scale real-world datasets demonstrates the model’s strong generalization ability. However, some experimental limitations hinder a comprehensive assessment of the model’s performance.

**Strengths:**

The incorporation of vision-language models is innovative, and experiments conducted on large-scale TSPLIB instances demonstrate the model’s strong generalization capability.

**Weaknesses:**

1.The main concern lies in the solution initialization process. The model uses LKH for initialization, which introduces bias in evaluating its standalone effectiveness. To more fairly demonstrate the model’s capability, the authors should consider using random or greedy initialization instead. Comparing a method initialized by LKH against other heuristics or neural solvers is inherently unfair. Moreover, since the proposed framework is initialized by LKH yet fails to consistently outperform LKH under comparable runtime conditions, its claimed effectiveness remains questionable.

2.Although the paper emphasizes scalability, it only evaluates 33 instances, which is insufficient to support the claim of large-scale generalization. The authors are encouraged to include TSPLIB instances with more than 500 nodes to better align with the paper’s stated focus on large-scale problems and to provide stronger empirical evidence.

3.Additional baselines should be incorporated for a more comprehensive comparison. Specifically, large-scale solvers such as RBG[1] and SIT[2] are relevant benchmarks. Furthermore, NeuroLKH[3], a neural variant based on LKH, should be included as it provides a closer reference point for hybrid approaches similar in spirit to the proposed method.

4.The abstract mentions that the proposed method reduces the optimality gap by 12% to 100%, but this claim is only briefly mentioned in the main text without sufficient explanation. Can the author further explain it?

[1] Hierarchically solving large-scale routing problems in logistic systems via reinforcement learning.
[2] Boosting Neural Combinatorial Optimization for Large-Scale Vehicle Routing Problems.
[3] Neurolkh: Combining deep learning model with lin-kernighan-helsgaun heuristic for solving the traveling salesman problem.

**Questions:**

See details in Weaknesses.

---

> ### Author Response · Authors · 2025-12-03
> **Response to Q1 - part a**
>
> We thank the Reviewer for the insightful comments. We respond to each comment as follows and hope our responses properly address your concerns.
>
> 1.	The main concern lies in the solution initialization process. The model uses LKH for initialization, which introduces bias in evaluating its standalone effectiveness. To more fairly demonstrate the model’s capability, the authors should consider using random or greedy initialization instead. Comparing a method initialized by LKH against other heuristics or neural solvers is inherently unfair. Moreover, since the proposed framework is initialized by LKH yet fails to consistently outperform LKH under comparable runtime conditions, its claimed effectiveness remains questionable.
>
> **Response**:
>
> **Clarifying key idea of ViTSP**. We do not introduce a new standalone trained solver, but rather demonstrate the generalizability and utility of VLM-guided decomposition heuristics to contribute to combinatorial optimization, while previous efforts of leveraging LLM/VLM, such as llm as optimizer and mTSP, suffer from delivering meaningful results even at small scale (n=50)
>
> **The rationale of initialization strategy selection**. We adopted LKH-3 for initialization because it is an open-source, easily accessible heuristic that has been commonly used for initialization in prior vehicle routing research, such as [1]. Our use of LKH is therefore aligned with existing practice and is not intended to introduce bias.
>
> **Fair comparison with aligned runtime.** Our study provided a fair comparison to showcase the effectiveness of ViTSP while in prior works, LKH-3 and Concorde were used only as a reference to avoid direct runtime-matched comparisons. To address the fairness concern, our original submission already includes an anytime comparison (Figures 2 & 5), where ViTSP and LKH-3 are evaluated under identical runtime budgets andthe  same initial solutions. This analysis isolates the effect of ViTSP’s VLM-guided decomposition from the choice of initializer and demonstrates that the proposed method consistently provides measurable improvements over simply running LKH-3 alone. To the best of our knowledge, this is the first comprehensive anytime comparison against LKH-3 and Concorde on large-scale TSPs and ViTSP is effectiveness in very-large-scale instances, while no single methods among LKH-3, Concorde, and ViTSP dominate the instance solving ranging between 1000 and 5000.
>
> [1] Learning to delegate for large-scale vehicle routing. NeurIPS 2021

---

> ### Author Response · Authors · 2025-12-03
> **Response to Q1 - part b**
>
> **Additional experiments using FI as initialization**. To explicitly examine the impacts of using different initialization, we have also tested initializing ViTSP with FI. We here report the average per-instance performance of ViTSP initialized by LKH (default) and FI, respectively, based on five runs under the same runtime budgets. **We have added this experiment in Section 4.3 to present the ablation results for different initialization methods.**
>
> | ViTSP             | dsj1000 | pr1002 | u1060 | vm1084 | pcb1173 | d1291 | rl1304 | rl1323 | nrw1379 | fl1400 | u1432 |
> |--------------------|---------|--------|-------|--------|---------|-------|--------|--------|---------|--------|--------|
> | runtime (s)        | 69.6    | 65.0   | 57.1  | 38.5   | 124.3   | 168.6 | 27.2   | 95.6   | 104.6   | 53.9   | 83.2   |
> | LKH initialization | 0.02%   | 0.01%  | 0.00% | 0.00%  | 0.16%   | 0.15% | 0.14%  | 0.04%  | 0.00%   | 0.18%  | 0.03%  |
> | FI initialization  | 0.50%   | 0.03%  | 0.64% | 0.67%  | 0.21%   | 2.42% | 8.01%  | 0.39%  | 0.01%   | 1.30%  | 1.43%  |
>
> | ViTSP             | fl1577 | d1655 | vm1748 | u1817 | rl1889 | d2103 | u2152 | u2319 | pr2392 | pcb3038 | fl3795 |
> |--------------------|--------|-------|--------|-------|--------|-------|--------|--------|---------|----------|---------|
> | runtime (s)        | 58.9   | 111.2 | 162.8  | 180.0 | 89.0   | 84.0  | 238.3  | 244.9  | 187.6   | 336.3    | 252.4   |
> | LKH initialization | 0.00%  | 0.00% | 0.05%  | 0.24% | 0.03%  | 0.39% | 0.08%  | 0.06%  | 0.09%   | 0.09%    | 0.57%   |
> | FI initialization | 17.61% | 6.56%  | 0.04% | 3.16%  | 0.50% | 23.55% | 0.98% | 0.07%  | 0.41%  | 0.34%   | 3.02%    |
>
> | ViTSP            | fnl4461 | rl5915 | rl5934 | pla7397 | rl11849 | usa13509 | brd14051 | d15112 | d18512 | pla33810 | pla85900 |
> |--------------------|---------|--------|--------|---------|----------|-----------|-----------|--------|--------|-----------|-----------|
> | runtime (s)        | 277.3   | 386.5  | 485.5  | 575.5   | 785.1    | 995.2     | 1725.3    | 2172.0 | 2319.0 | 5758.1    | 29863.6   |
> | LKH initialization | 0.16%   | 1.13%  | 0.41%  | 0.28%   | 1.03%    | 0.47%     | 0.31%     | 0.22%  | 0.36%  | 0.52%     | 0.83%     |
> | FI initialization  |  2.49%  |8.42%   | 6.40%  | 5.72%  | 9.36%   | 8.89%    | 0.52%     | 0.24%     | 0.54%  | 10.65% | 2.81%     |
>
> The ViTSP initialized by LKH consistently outperforms the one initialized by FI. These results confirm that while initialization affects performance, as expected, the VLM-guided decomposition consistently improves over its corresponding initialization baseline.
>
> Overall, our experiments show that ViTSP provides significant gains over LKH-3 on very-large-scale instances (n>5000), where the VLM-guided decomposition is most beneficial. For smaller instances (1,000–2,000 nodes), pure LKH-3 or Concorde remains state-of-the-art, which is consistent with known results in the literature. Importantly, while no single model can surpass LKH-3 in all regimes (as we show NeuroLKH results in response to Q3 below), our work introduces a new general-purpose hybrid paradigm that integrates VLM priors with classical OR heuristics to achieve superior and generalizable performance, a direction that is orthogonal and complementary to existing solvers.

---

> ### Author Response · Authors · 2025-12-03
> **Response to Q2 and Q3**
>
> 2.	Although the paper emphasizes scalability, it only evaluates 33 instances, which is insufficient to support the claim of large-scale generalization. The authors are encouraged to include TSPLIB instances with more than 500 nodes to better align with the paper’s stated focus on large-scale problems and to provide stronger empirical evidence.
>
> **Response**:
>
> Thank you for highlighting the scalability in the evaluation. Our experimental design was explicitly constructed with scalability as a primary focus, as stated in Section 4.1, lines 308-310 of the manuscript:
>
> > We chose TSP instances with N ≥ 1, 000 from the dataset to represent (very-)large-scale problems, where exact solvers be-
> gin to struggle.
>
> All of these instances are actually large-scale and we provided instance-specific performance comparison.  We believe this has already provided diverse benchmark instances for large-scale generalization. For example, the test set spans instances such as pr1002 (1,002 nodes) up to pla85900 (85,900 nodes), thereby covering from 1k to 80k in problem size.
>
> ***
> 3.	Additional baselines should be incorporated for a more comprehensive comparison. Specifically, large-scale solvers such as RBG[1] and SIT[2] are relevant benchmarks. Furthermore, NeuroLKH[3], a neural variant based on LKH, should be included as it provides a closer reference point for hybrid approaches similar in spirit to the proposed method.
>
> **Response**:
>
> Thank you for providing additional benchmark algorithms. **Our selection followed the spirit of benchmarks used in prior work.**  Despite that neither the baselines employed in our study nor related methods such as RBG and SIT have originally included NeuroLKH as a baseline, we have added additional experiments for SIT[2] and NeuroLKH[3] below to demonstrate the performance comparison. We are unable to add RBG[1] because it did not release open-source codes for reproduction.
>
> The runtimes of SIT and NeuroLKH are aligned with ViTSP for fair comparison.  The results show that SIT reaches an optimality gap ranging from 1% -5%, generally less performant than ViTSP.  We agree that NeuroLKH is a strong benchmark as it reaches very small gaps. However, it actually does not surpass every instance for 33 TSPLIB, as we highlight the best result in the table above.
>
>
> | Method       | dsj1000 | pr1002 | u1060 | vm1084 | pcb1173 | d1291 | rl1304 | rl1323 | nrw1379 | fl1400 | u1432 |
> |--------------|---------|--------|--------|--------|---------|--------|--------|--------|---------|--------|--------|
> | Runtime (s)  | 69.6    | 65.0   | 57.1  | 38.5   | 124.3   | 168.6 | 27.2   | 95.6   | 104.6   | 53.9   | 83.2   |
> | NeuroLKH     | **0.00%**  | **0.00%** | 0.00%  | 0.06%  | **0.05%**   | **0.08%**  | 0.19%  | **0.01%**  | 0.00%   | 0.18%  | **0.00%**  |
> | SIT          | 1.21%   | 1.13%  | 1.33%  | 0.83%  | 1.54%   | 4.33%  | 1.51%  | 1.35%  | 1.55%   | 4.52%  | 2.28%  |
> | ViTSP (LKH)  | 0.02%   | 0.01%  | **0.00%**  | **0.00%**  | 0.16%   | 0.15%  | **0.14%**  | 0.04%  | **0.00%**   | **0.18%**  | 0.03%  | |
>
> | Method       | fl1577 | d1655 | vm1748 | u1817 | rl1889 | d2103 | u2152 | u2319 | pr2392 | pcb3038 | fl3795 |
> |--------------|--------|--------|--------|--------|--------|--------|--------|--------|---------|----------|--------|
> | Runtime (s)  | 58.9   | 111.2 | 162.8  | 180.0 | 89.0   | 84.0   | 238.3  | 244.9  | 187.6   | 336.3    | 252.4  |
> | NeuroLKH     | 0.06%  | 0.00% | **0.03%**  | **0.13%** | 0.04%  | **0.00%** | 0.08%  | **0.03%** | **0.01%**   | **0.01%**    | **0.50%**  |
> | SIT          | 5.04%  | 5.36% | 1.17%  | 2.28% | 4.01%  | 5.60%  | 4.21%  | 0.22%  | 1.12%   | 1.95%    | 7.55%  |
> | ViTSP (LKH)  | **0.00%**  | **0.00%** | 0.05%  | 0.24% | **0.03%**  | 0.39%  | **0.08%**  | 0.06%  | 0.09%   | 0.09%    | 0.57%  |
>
> | Method       | fnl4461 | rl5915 | rl5934 | pla7397 | rl11849 | usa13509 | brd14051 | d15112 | d18512 | pla33810 | pla85900 |
> |--------------|----------|--------|--------|----------|----------|-----------|-----------|--------|--------|-----------|-----------|
> | Runtime (s)  | 277.3    | 386.5  | 485.5  | 575.5    | 785.1    | 995.2     | 1725.3    | 2172.0 | 2319.0 | 5758.1    | 29863.6   |
> | NeuroLKH     | **0.07%**    | **0.01%**  | **0.02%**  | **0.04%**   | **0.88%**    | **0.01%**     | 0.41%     | 0.31%  | **0.32%**  | **0.20%**     | **0.58%**     |
> | SIT          | 1.42%    | 3.46%  | 5.09%  | 1.96%    | 5.01%    | 12.34%    | 4.39%     | 3.30%  | 3.73%  | 4.40%     | 6.60%     |
> | ViTSP (LKH)  | 0.16%    | 1.13%  | 0.41%  | 0.28%    | 1.03%    | 0.47%     | **0.31%**     | **0.22%**  | 0.36%  | 0.52%     | 0.83%     |
>
>
> [1] Hierarchically solving large-scale routing problems in logistic systems via reinforcement learning.
>
> [2] Boosting Neural Combinatorial Optimization for Large-Scale Vehicle Routing Problems.
>
> [3] Neurolkh: Combining deep learning model with lin-kernighan-helsgaun heuristic for solving the traveling salesman problem.

---

> ### Author Response · Authors · 2025-12-03
> **Response to Q4**
>
> 4.	The abstract mentions that the proposed method reduces the optimality gap by 12% to 100%, but this claim is only briefly mentioned in the main text without sufficient explanation. Can the author further explain it?
>
> **Response**:
>
> Thank you for the question. The reported “12% to 100% reduction in optimality gap” refers to a relative improvement of ViTSP over LKH-3. In our experiments on 33 TSPLIB instances, we allocated the identical wall-clock time to both LKH-3 and ViTSP. Under this fair comparison, ViTSP achieves smaller optimality gaps in 20 out of 33 instances. This relative gap reduction is calculated as (LKH-3’s gap – ViTSP’s gap) / (LKH-3’s gap). Across the 20 instances where ViTSP performs better, the relative gap reduction ranges from 12% to 100%. A 100% reduction indicates that ViTSP reaches global optimality within the allotted time while LKH-3 does not make comparable progress. We have modified lines 366-368 to elaborate this result.

---

### Author Response · Authors · 2025-12-03
**General Responses**

Dear reviewers, ACs and SACs,

Thank you for your review and efforts. We summarize our key contributions below. We have also highlighted changes in blue in the manuscript. We hope our responses below resolve your concerns. If so, we would be grateful if you would consider updating your score.

In this study, we propose a novel VLM-based approach for solving large-scale TSPs (ViTSP). The core contributions that distinguish our work from prior studies are:

(1)	**Multimodal decomposition heuristics from VLMs without dedicated training**. Prior neural TSP heuristics or solvers require extensive learning. In contrast, ViTSP shows, for the first time to our knowledge, that a pretrained VLM can interpret large-scale 2D spatial layouts and identify promising subregions for iterative improvements without any task-specific training, which is fundamentally different from existing neural methods.

(2)	**A new hybrid generative AI-OR pipeline enabling solving large problems where neural solvers struggle**. ViTSP fuses the advantage of pre-trained VLMs and off-the-shelf solvers that offers a previously unexplored path for scaling TSP in unseen instances.  Considering their computational overhead, an asynchronous coordination between VLM and the exact solver is developed to further accelerate this process for practical use. In contrast, prior LLM/VLM approaches directly construct solutions with poor quality, which limits practical performance at scale. Neural solvers often show poor performance in out-of-distribution instances.

(3)	**Fair experimental comparison among benchmarks to demonstrate ViTSP’s performance**. We selected baselines because they are widely recognized for their superior performance. However, prior works (e.g., SIT, INViT, UDC) did not consistently align runtime budgets when reporting results. LKH-3 was used only as a reference to avoid direct runtime-matched comparisons; yet when LKH-3 is given the same runtime as these learning-based models, it can achieve smaller optimality gaps, even if not fully optimal. In our study, we have put substantial efforts to make strict comparison across benchmarks by aligning runtime, especially well-known Concorde and LKH-3. We further provide instance-specific evaluations to assess performance on out-of-distribution TSPLIB instances. Specifically, with aligned runtime budgets, ViTSP outperforms LKH-3 by reducing its gaps by 12% to 100% and consistently surpasses existing learning-based approaches.

We believe this work opens a new perspective on fusing generative AI with operations research to address combinatorial optimization problems at a practical scale.

---

### Meta-Review · Area_Chair_urNQ · 2025-12-24

**Summary:**

The reviewers recognize the innovative incorporation of a VLM in hybrid method. They asked clarifications questions (e.g., runtime measurement) and mentioned the initialization with LKH, the empirical evaluation on limited instances, the lack of relevant baselines, the lack of sensitivity analysis of VLM output, the lack of theoretical analysis and discussion, or the extension to other problems than TSP.

**Reviewer Concerns:**

To address the concerns, the authors provided several additional experimental results:
- Regarding the initialization with LKH, the authors demonstrate the overall approach can still work with another heuristic initialization, albeit with lower performance. Also, I believe this initialization is acceptable since the computation time of the initialization is taken into account.
- Regarding the limited instances used for the evaluation, the authors confirm the effectiveness of their approach on random large TSP instances (as done in previous work).
- Regarding baselines, the authors demonstrate that their approach is competitive, notably with NeuroLKH.
In the rebuttal, the authors discussed how to extend their approach to CVRP, which is fine in my opinion.

Two concerns are still outstanding: the sensitivity analysis of VLM output and the lack of theoretical analysis. A discussion of those points would have certainly strengthened the paper.

**Reviewer Scores:**

Reviewer AY9B and Reviewer ZmWq could have raised their score since their criticisms are mostly addressed in the rebuttal.

Reviewer RptF would have probably kept their score, since the two outstanding concerns were raised by this reviewer.

Reviewer v67N would also have probably kept their score.

---

### Decision · Program_Chairs · 2026-01-26

Accept (Poster)